# Untangling tradeoffs between recurrence and self-attention in neural networks

**Giancarlo Kerg**[1,2,*]   **Bhargav Kanuparthi** [1,2,*]   **Anirudh Goyal** [1,2]   **Kyle Goyette** [1,2,3]

**Yoshua Bengio**[1,2,4]                    **Guillaume Lajoie**[1,2,5]

## Abstract

Attention and self-attention mechanisms, are now central to state-of-the-art deep learning on sequential tasks. However, most recent progress hinges on heuristic approaches with limited understanding of attention's role in model optimization and computation, and rely on considerable memory and computational resources that scale poorly. In this work, we present a formal analysis of how self-attention affects gradient propagation in recurrent networks, and prove that it mitigates the problem of vanishing gradients when trying to capture long-term dependencies by establishing concrete bounds for gradient norms. Building on these results, we propose a relevancy screening mechanism, inspired by the cognitive process of memory consolidation, that allows for a scalable use of sparse self-attention with recurrence. While providing guarantees to avoid vanishing gradients, we use simple numerical experiments to demonstrate the tradeoffs in performance and computational resources by efficiently balancing attention and recurrence. Based on our results, we propose a concrete direction of research to improve scalability of attentive networks.

## 1   Introduction

We live in a world where most of the information takes a sequential form, largely because it is delivered over time. Performing computations on streams of sequential inputs requires extracting relevant temporal dependencies and learning to recognize patterns across several timescales. Humans can effortlessly make associations relating events stored in memory which are far from each other in time and thus, capture long-term dependencies.

Historically, recurrent neural networks (RNNs) have been the deep network architecture of choice for this type of task since, just like neural circuits in the brain, they enable *dynamics* that can be shaped to interact with input streams. However, RNNs (including gated RNNs [35, 10]) still struggle with large timescales as their iterative nature leads to unstable information propagation [5, 30, 35, 18].This is because most standard RNNs rely on their current state $h_t$, a vector of fixed dimension, to represent a summary of relevant past information. Indeed, Bengio et al. [5] showed that without making additional assumptions, storing information in a fixed-size state vector in a stable way necessarily leads to vanishing gradients when back-propagating through time (see also [18]).

---

[*]Indicates first authors. Ordering determined by coin flip.
1: Mila - Quebec AI Institute, Canada
2: Université de Montréal, Département d'Informatique et Recherche Opérationelle, Montreal, Canada
3: Université de Montréal, CIRRELT, Montreal, Canada
4: CIFAR senior fellow
5: Université de Montréal, Département de Mathématiques et Statistiques, Montreal, Canada

Correspondence to: <giancarlo.kerg@gmail.com>

Several attempts have been made to augment RNN dynamics with external memory to mitigate these issues [37, 14, 34, 15], but it is only recently that access to externally stored information has become effective with the introduction of *attention*, and more particularly *soft attention* mechanisms [4]. Attention provides a way by which a system can dynamically access past states and inputs across several timescales, bypassing the need of sequential propagation and ignoring irrelevant information (or distractor information). There is substantial empirical evidence that attention, especially *self-attention* (Vaswani et al. [38], Ke et al. [22]), is very helpful to improve learning and computations over long-term dependencies. However, to the best of our knowledge, there is currently limited understanding of gradient scaling properties in the presence of attention. Moreover, attending over long sequences requires to hold inputs and/or past states in memory, a process that typically scales quadratically with sequence length.

Much like work from the '90s established formal results for gradient exploding/vanishing in deep/recurrent networks [5], we believe it is crucial to establish similar theoretical tools for attention mechanisms, as these methods are under intense development where scalability and complexity are important issues. In this paper, we contribute to this direction with a *formal analysis of gradient propagation in self-attentive systems which precisely quantify trade-offs between recurrence and attention*, offering valuable guarantees for attention mechanism development. Concretely exploiting these theorems, we propose a simple family of screening mechanisms to *maximally reduce computational complexity and memory usage, while simultaneously maintaining good gradient propagation over large time scales*. Using simple tasks for their ease of interpretation, and their variety of computational demands, we illustrate the efficacy of this approach in numerical experiments.

The remainder of this paper is as follows. In Section 2, we give a brief outline of related cognitive processes and neural network mechanisms. In Section 3, we present our central results: asymptotic guarantees for gradient propagation in self-attentive recurrent networks. To illustrate how to exploit these guarantees, in Section 4, we showcase a simple *relevancy screening mechanism* that aims to efficiently consolidate relevant memory, reducing the size of the computational graph from quadratic to linear in sequence length. Finally, in Section 5, we compare various recurrent and attention models with our proposed relevancy screening mechanism on a series of simple numerical experiments, while, in Section 6, we analyze their gradient propagation properties together with their GPU usage.

## 2 Background

To perform complex tasks, our brains rely on mechanisms to encode and retrieve information to and from memory [40, 33]. In contrast, standard RNNs follow rigid sequential dynamics as they are parametric i.e with a fixed-size state vector. Self-attention methods can overcome this limitation by giving access to previous past states for computing the next state. For the sake of the discussion, we call such RNNs, which are augmented by the memory of past states as *semi-parametric* RNNs. The use of soft-attention [4] in such models has improved performance on many tasks such as reading comprehension, abstractive summarization, textual entailment and learning task-independent sentence representations [29, 26, 31, 39] as well as in the self-supervised training of extremely large language models [12, 32] due to their ability to handle long-term dependencies.

Intriguingly, the most notable advances in the use of attention is in purely attention-based systems such as the Transformer [38], which completely foregoes recurrence and inspired some of the work listed above. While the performance of these systems is impressive, their memory and computation requirements grows quadratically with the total sequence length. To address this issue, many variants that aim to "sparsify" the attention matrix have been proposed. Notably, Ke et al. [22] developed the Sparse Attentive Backtracking model (SAB), a self-attentive Long Short-Term Memory network (LSTM) [35] that leverages sparsity by selecting only the top-$k$ states in memory based on an attention score, propagating gradients only to those chosen hidden states. Recently, Zhao et al. [41] propose to use a similar top-$k$ attention, and Child et al. [9] introduce sparse masks which attends to roughly $\sqrt{n}$ locations in memory, implementing explicit selection methods for Transformers. Reformer models [23] replace the dot-product attention by locality-sensitive hashing, changing its complexity from $O(T^2)$ to $O(T)$, where $T$ is the sequence length. Finally, TransformerXL [11] enables learning dependencies beyond a fixed length without disrupting temporal coherence and has resulted in state of the art performance in language models.

Still, most of these approaches naively sub-sample input streams for memory storage. Our brains on the other hand, seem to select relevant information from the recent past to commit to long term memory based on their relevancy, a process often referred to as memory consolidation [1]. Attempts at mimicking this sparse temporal selectivity process has shown great promise in a variety of contexts [14, 28, 16, 13], and our work aims to formalize this idea for self-attentive recurrent networks.

# 3 Theoretical analysis of gradient propagation

In this section, we analyze the influence of self-attention onto gradient propagation in recurrent networks with self-attention. In order to do so let us first recall the equations of a recurrent neural network with self-attention. We note that even though we are using "vanilla RNNs" in the formulations of our results, any recurrent network can take its place (see Section 5 where we use LSTMs in the experiments). Let $x_t \in \mathbb{R}^m$ be the input and $h_t \in \mathbb{R}^n$ be the hidden state at time step $t$, satisfying the update equation for all $t \geq 1$,

$$h_{t+1} = \phi(Vs_t + Ux_{t+1} + b) \tag{1}$$
$$s_t = f(h_t, c_t) \tag{2}$$

where $\phi$ is a non-linearity, $f : \mathbb{R}^n \times \mathbb{R}^n \to \mathbb{R}^n$, $V \in \mathbb{R}^{n \times n}$, $U \in \mathbb{R}^{n \times m}$, $b \in \mathbb{R}^n$ and $c_t = \alpha_{1,t}h_1 + \alpha_{2,t}h_2 + \ldots + \alpha_{t,t}h_t$ with $\alpha_{i,t} := \frac{\exp(e_{i,t})}{\sum_{j=1}^{t} \exp(e_{j,t})}$ and $e_{i,t} := a(s_{t-1}, h_i)$, where $a : \mathbb{R}^n \times \mathbb{R}^n \to \mathbb{R}^n$ is the attention *alignment function*. Throughout, we assume training is done via gradient descent of a cost function $L$ using back-propagation.

Oftentimes, one uses $s_t = f(h_t, c_t) = h_t + c_t$ (but concatenation would be more general), and for all $t > 1$ and $1 \leq j \leq t$, $a(s_{t-1}, h_j) = v_a^T \cdot \tanh(W_a \cdot s_{t-1} + U_a \cdot h_j)$, where $v_a \in \mathbb{R}^n$, and $W_a, U_a \in \mathbb{R}^{n \times n}$. The latter choice for alignment function is sometimes referred to as "additive self-attention" and was used in the original paper [4]. We emphasize that the results presented in this section hold independently of the choice of the alignment function as, we will discuss later in this section. Lastly, while results presented below are relatively succinct, their derivations are involved and we refer the interested reader to the Appendix for detailed proofs.

## 3.1 Preliminaries

Our goal in this section is to establish formal propagation rules for a system where multiple paths of signal propagation are possible. We would like to understand the relationship between skip connections (those coming from self-attention) and recurrent connections, as well as how the interplay between the two leads to good gradient propagation. In order to achieve this, we seek to analyze the asymptotic behaviour of $\|\nabla_{h_t} L\| = \|\left(\frac{ds_T}{dh_t}\right)^T \nabla_{s_T} L\|$, as $T \to \infty$. We accomplish this by decomposing $\nabla_{h_t} L$ with respect to all possible gradient backpropagation paths, or in other words, by decomposing $\frac{ds_T}{dh_t}$ into sums of products of Jacobian matrices corresponding to those gradient paths, using Proposition 1.

**Proposition 1.** *For all $t \geq 1$, $k \geq j \geq 0$, $k' \geq 0$, let $E_{k'}^{(t)} = \frac{\partial s_{t+k'}}{\partial h_t}$, and $F_{k+1,j}^{(t)} = \frac{\partial s_{t+k+1}}{\partial h_{t+j+1}} \cdot J_{t+j} + 1_{j=k} \cdot \frac{\partial s_{t+k+1}}{\partial s_{t+k}}$, with $J_{t+j}$ the Jacobian matrix $\frac{dh_{t+j+1}}{ds_{t+j}}$. Then, we have*

$$\frac{ds_{t+k}}{dh_t} = \sum_{s=0}^{k} \bar{\xi}_{0:k}^{(t)}(s) \tag{3}$$

*where for all $s \geq 1$, $\bar{\xi}_{0:k}^{(t)}(s) = \sum_{0 \leq i_1 < \ldots < i_s < k} F_{k,i_s}^{(t)} \cdot F_{i_s,i_{s-1}}^{(t)} \cdot \ldots \cdot F_{i_2,i_1}^{(t)} \cdot E_{i_1}^{(t)}$ and where $\bar{\xi}_{0:k}^{(t)}(0) = E_k^{(t)}$. (Proof in Appendix A.2, Proposition 1)*

Here, each term $F_{k,i_s}^{(t)} \cdot F_{i_s,i_{s-1}}^{(t)} \cdot \ldots \cdot F_{i_2,i_1}^{(t)} \cdot E_{i_1}^{(t)}$ corresponds to exactly one gradient path involving exactly $s + 1$ skip connections going from $t$ to $t + k$, via the $s$ hidden states $h_{t+i_s+1}, \ldots, h_{t+i_1+1}$. In particular, $\bar{\xi}_{0:k}^{(t)}(s)$ is the sum of all terms containing exactly $s$ Jacobian matrices $J$, and thus the larger $s$ is, the more $\bar{\xi}_{0:k}^{(t)}(s)$ is prone to vanishing.

**Intuition:** In order to find paths that are not vanishing as $T \to \infty$, we want to find gradient paths with: **(i)** a bounded path length $s$ so that the number of Jacobian matrices involved in the product is limited. **(ii)** attention scores that are sufficiently bounded away from 0, so that the resulting product of attention scores is sufficiently bounded away from 0 as well. In order to see how exactly the attention weights come into play via matrices $E$ and $F$, we refer to Proposition 2 from Appendix A.2.

**Defintions:** Let us fix an integer $t \geq 1$, an integer $s \in \{1, 2, \ldots, T-t\}$, and an ordered set of indices $i_1, i_2, \ldots, i_s \in \{0, 1, \ldots, T-t-1\}$, verifying $i_1 \leq i_2 \leq \ldots \leq i_s$.

- For sequences $\{g(T)\}_{T \geq 1}$ and $\{f(T)\}_{T \geq 1}$, we say that $\underline{f(T) = \Omega(g(T))}$ if there exists positive constants $c$ and $T_0$ such that $f(T) \geq c \cdot g(T)$ for all $T \geq T_0$.
- At time $t$, we call a past hidden state $h_i$ a *relevant event* if the attention weight $\alpha_{i,t}$ is sufficiently bounded away from zero.
- We call the $s$-tuple $(i_1, i_2, \ldots, i_s)$ a *dependency chain $\gamma$ of depth $s$*, as it induces a gradient backpropagation path going via the $s$ hidden states $h_{t+i_s+1}, \ldots, h_{t+i_1+1}$.
- We call *dependency depth* the *smallest* depth among all dependency chains where the product of the corresponding attention scores is $\Omega(1)$ as $T \to \infty$.

The central message is that *if the dependency depth is bounded from above and sufficiently small, then we mitigate gradient vanishing*. As we see below, task structure introduces different ways in which this may happen. We now present a formal treatment for specific cases, and lay the groundwork to take advantage of this structure during learning.

## 3.2 Uniform relevance case

Suppose each state has equal relevance in some task. What can be said about gradient propagation? This translates to having each attention weight $\alpha_{i,t} = 1/t$ for all $t \geq i \geq 1$. We then have dependency chains of depth 1 but with vanishing rate $\Omega(1/T)$, as formalized in the following theorem (cf. A.3)

**Theorem 1.** *Let $h_t$ be the hidden state at time $t$ of a vanilla RNN with uniform attention, under mild assumptions on the connectivity matrix $V$, and trained with respect to a loss L, then if $T$ is the total sequence length, we have*

$$\|\nabla_{h_t} L\| = \Omega(1/T) \tag{4}$$

*as $T \to \infty$. (proof in Appendix A.3, Theorem 1)*

This corresponds to the case where all past events contribute equally error signals. We also note that this result is independent of the choice of the alignment function $a$ (cf. Remark 8 in the Appendix A.3).

**Intuition:** As a "worst case scenario" Theorem 1 reveals the true trade-off of early self-attentive recurrent networks [4]. On one hand, the lower bound obtained on gradient norm is substantially better than in a vanilla RNN without attention, where vanishing happens at an exponential rate, as opposed to a polynomial one here. This situation does not lend itself to sparse memory approaches as all events need to be held in memory, thus conserving quadratically scaling complexity. In contrast many inputs and tasks do not call for uniform attention and naturally lend themselves to sparse dependency paths for computation. The next case treats this situation. Nevertheless, this uniform attention bound is applicable in practice for two reasons: (1) typically, attention weights are initialized uniformly, and early training may result in gradients best described by this regime. (2) We experimentally verified that gradient propagation remains stable throughout training for a fully self-attentive RNN, where this bound is relevant, see Fig 2 (Section 6).

## 3.3 Sparse relevance case with bounded dependency depth

Now let us look at a more realistic case where only a sparse subset of past states are relevant for the task at hand, and the gradient needs to access those states efficiently for good learning. Figure 1 illustrates this scenario by showing the attention scores for two input examples computed by a simple self-attentive model [4], trained on Copy and Denoise tasks respectively (see Section 5). This structure introduces the possibility to impose sparsity in the computational graph, and to limit memory use. With these constraints in mind, the goal is to engineer dependency chains that enable best gradient propagation between these relevant events.

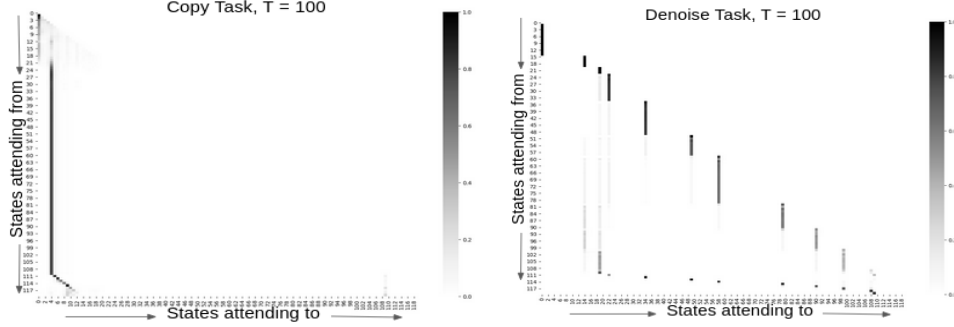

Figure 1: *Magnitude of attention weights between states in a trained, fully recurrent and fully attentive model* (Bahdanau et al. [4]). Each pixel in the lower triangle corresponds to the attention weight of the skip connection departing from the the state marked on the $y$-axis to the state marked on the $x$-axis. Left shows Copy task, right shows Denoise task. Task details in Section 5

**Notation**: We consider a $\kappa$-sparse attention mechanism of dependency depth $d$.

- *Sparsity coefficient*: $\kappa \geq 1$. Borrowing from the SAB model [22], at each time step, attention is allowed at most $\kappa$ relevant events from the past. That is, for any $t$ there are at most $\kappa$ indices $i$ such that $\alpha_{i,t} \neq 0$, which gives rise to a sparse temporal segmentation via the most relevant events.

- *Maximal dependency depth*: $d$. This is the maximal dependency depth across all time steps $t$.

**Theorem 2.** *Let $h_t$ be the hidden state at time $t$ of a vanilla RNN with $\kappa$-sparse uniform attention mechanism of maximal dependency depth $d$, and under mild assumptions on the connectivity matrix $V$, then*

$$\|\nabla_{h_t} L\| = \Omega(1/\kappa^d) \tag{5}$$

*as $T \to \infty$. (proof in Appendix A.4, Theorem 2)*

Similarly to uniform case, Theorem 2 is independent of the choice of the alignment function $a$ (cf. remark 19 in the appendix).

**Intuition:** Notice the dependency depth $d$ affects the lower bound exponentially, while $\kappa$ affects it polynomially. In other words, the number of relevant events attended to at each time step contributes far less to gradient vanishing than the number of events in the longest dependency chain. Theorem 2 outlines the tradeoff between computational complexity as $T \to \infty$ and gradient propagation when balancing attention and recurrence. Attending directly to many relevant past events reduces $d$ and ensures good gradients at the expense of the complexity cost associated with storing past events and computing attention scores (the strategy employed by Transformers [38]). On the other hand, enforcing small sparsity coefficient $\kappa$ helps keep computational complexity low ($O(\kappa T)$), but forces the error gradient through recurrent paths, thereby augmenting the dependency depth $d$ and degrading gradient signal strength. Importantly, $\kappa$ and $d$ co-vary in ways that depend on the task's underlying relevancy structure, a point that is explained in detail in Appendix C (See Fig 3). In the extreme case where $\kappa$ and $d$ are assumed to be bounded, we have $\Omega(1/\kappa^d) = \Omega(1)$, and thus we mitigate gradient vanishing. In other situations where $\kappa$ and $d$ scale in other ways, an explicit sparsification strategy can be derived by exploiting Theorem 2, as we illustrate in the next section.

## 4 Relevancy screening mechanism

Equipped with the results from the previous section, we wish to refine heuristics that strike a balance between good gradient propagation and computational/memory complexity. Building on the SAB model [22], we remark that although sparse attention attends to the top-$\kappa$ events at any point in time, attention scores must be computed on all events stored in memory to extract the $\kappa$ best ones. Thus, the resource bottleneck is not controlled by $\kappa$, but rather by the number of stored events in memory. In SAB, there is a naive attempt to control this number by only recording network states at each 10

time steps. However, this reduces the size of the computational graph only by a constant factor, but retains $O(T^2)$ complexity. In contrast, Theorem 2 tells us that the only important events to conserve for good gradient propagation are the *relevant* ones (also see Remark 6 in Appendix A.2). Thus, we propose to reduce complexity while maintaining good gradient propagation by selectively storing events that are predicted to be relevant in the future, using a *relevancy screening mechanism*.

---

**Algorithm 1** Relevancy Screening

---

1: **procedure:** RelRNN($\mathbf{s}_{t-1}, \mathbf{x}_t$)
    **Require:** Previous macro-state - $\mathbf{s}_{t-1}$
    **Require:** Input - $\mathbf{x}_t$, $\nu > 0$, $\rho > 0$
    **Require:** Short-term buffer $s_{t-1}^{(i)} \in S_{t-1}$
    **Require:** Relevant set $r_{t-1}^{(i)} \in R_{t-1}$
2:   $h_t \leftarrow \phi(V\mathbf{s}_{t-1} + U\mathbf{x}_t + b)$
3:   $S_t = S_{t-1}.\text{add}(h_t)$
4:   **if** $t - \nu > 0$ **then**
5:      $S_t = S_t.\text{remove}(h_{t-\nu})$
6:   **if** $t - \rho > 0$ **and** $C(t-\rho) = True$ **then**
7:      $R_t = R_{t-1}.\text{replaceWith}(h_{t-\rho})$
8:   $M_t = [S_t, R_t]$
9:   **for all** $m^{(i)} \in M_t$ **do**
10:     $\tilde{z}^{(i)} \leftarrow v_a^T \cdot \tanh\left(W_a\mathbf{s}_{t-1} + U_a m^{(i)}\right)$
11:   $z \leftarrow \text{softmax}(\tilde{z})$
12:   $\mathbf{s}_t = h_t + \sum_i z^{(i)} m^{(i)}$
13:   **return** $\mathbf{s}_t$

---

The idea is simple: devise a screening function $C(i)$ which estimates the future relevance of $h_i$, and store selected events in a *relevant set* $R_t = \{h_i | i < t \wedge C(i) = True\}$ for future attention. In principle, one can explicitly control how $R_t$ grows with $t$, thus mitigating the complexity scaling outlined above. Here, $C(i)$ could take many forms, the best of which depends on task structure. In what follows, we present an example screening mechanism meant to showcase the lessons learned from Theorem 2, but we refer the interested reader to Section 7 for further possibilities.

We take inspiration from memory consolidation principles in human cognition [1], which defines the transfer of events from short-term to long-term memory. We remark that for some tasks such as those depicted in Figure 1, relevance varies very little across time. To implement relevancy screening for such tasks, at every time step $t$ we attend to two subsets of the past hidden states. We call the first subset a *short-term buffer* $S_t = \{h_{t-\nu}, h_{t-\nu+1}, .., h_{t-1}\}$ which consists of the hidden states of the last $\nu$ time steps, while the second subset is the relevant set $R_t$. We compute the *relevance score* at time step $i$, $\beta(i) = \sum_{j=i}^{i+\nu-1} \alpha_{i,j}$, measuring the integrated attention scores over our short-term buffer $S_t$. More precisely, $C(i)$ is satisfied if $\beta(i)$ is part of the top $\rho$ relevance scores when compared to all previously observed hidden states, where $\rho$ is a fixed hyper-parameter satisfying $\rho \geq |R_t|$ for all $t$. The pseudo-code in Algorithm 1 describes the screening mechanisms and the interaction between the short-term buffer $S_t$ and a finite size relevant set $R_t$. '.replaceWith()' is a function replacing the hidden state with the lowest relevance score by the hidden state in the argument.

To see how the relevancy screening mechanism is grounded in the theory developed in Section 3, note that the sets $S_t$ and $R_t$ give rise to a sparse attention mechanism with sparsity coefficient $\kappa$ satisfying $\kappa = \nu + \rho \geq |S_t| + |R_t|$. Hence, memory complexity is constant while the $O(T^2)$ bottleneck of computational complexity is replaced by $O((\rho+\nu)\cdot T) = O(T)$. Lastly, applying Theorem 2, we get the following guarantee for all $t \geq 0$: $\|\nabla_{h_t} L\| = \Omega(1/(\rho+\nu)^d)$ as $T \to \infty$. Thus the choices of $\nu$ and $\rho$ not only directly impact computational complexity and gradient propagation, but also indirectly influence gradient propagation via the implicit effect of $\kappa = \nu + \rho$ on $d$ as already discussed in Section 3. Finally, as already mentioned, see Fig 3 in Appendix C, where we perform an experimental trade-off analysis between $\kappa$ and $d$ by tweaking $\rho$ and $\nu$ in the relevancy screening mechanism.

## 5 Experiments

Before describing experiments, we make a few remarks. First, we stress that Relevancy Screening can be applied to any semi-parametric attentive model but we refer to the version presented below, which uses an RNN/LSTM base, as RelRNN/RelLSTM ("*Relevance RNN /LSTM*"). Second, our objective is not to find state-of-the-art performance but to highlight the advantages of event relevancy and selective sparsity. Finally, we note that relevancy-based sparsity does not necessarily improve performance over fully attentive models, but rather allows efficient and scalable usage. As we show below, RelRNN and RelLSTM perform almost identically to other self-attentive recurrent models (e.g. [4, 22]) on simple tasks, but use considerably less memory and compute complexity. In what

Table 1: Results for Transfer Copy task.

| $T$ | 100 | 200 | 400 | 2000 | 5000 |
|---|---|---|---|---|---|
| orth-RNN | 99% | 4% | 16% | 10% | 0% |
| expRNN | 100% | 86% | 73% | 58% | 50% |
| MemRNN | 99% | 99% | 99% | 92% | OOM |
| RelRNN | **100%** | 99% | 99% | 99% | 99% |
| LSTM | 99% | 64% | 48% | 19% | 14% |
| h-detach | 100% | 91% | 77% | 51% | 42% |
| SAB | 99% | 95% | 95% | 95% | 95% |
| RelLSTM | **100%** | 99% | 99% | 99% | 99% |

Table 2: Results for Denoise task.

| $T$ | 100 | 300 | 500 | 1000 | 2000 |
|---|---|---|---|---|---|
| orth-RNN | 90% | 71% | 61% | 29% | 3% |
| expRNN | 34% | 25% | 20% | 16% | 11% |
| MemRNN | 99% | 99% | 99% | 99% | OOM |
| RelRNN | **100%** | 99% | 99% | 99% | 99% |
| LSTM | 82% | 41% | 33% | 21% | 15% |
| GORU | 92% | 93% | 91% | 93% | 73% |
| SAB | 99% | 99% | 99% | 99% | 99% |
| RelLSTM | **100%** | 99% | 99% | 99% | 99% |

follows, we denote MemRNN/MemLSTM for vanilla self-attention RNN/LSTM as defined in [4]. We also refer to Appendices C, B, D for additional experimental results and implementation details.

## 5.1 Tasks with sparse dependency chains

A good stereotypical task type that captures sparse sequences of important events are memorization tasks. Here, the network has to memorize a sequence of relevant characters presented among several non-relevant ones, store it for some given time delay and output them in the same order as they were read towards the end of the sequence.

**Copy task** [19]: The characters to be copied are presented in the first 10 time steps, then must be outputted after a long delay of $T$ time steps (see full description in Arjovsky et al. [2]). Thus, all the *relevant events* occur in the first 10 time steps. This can be corroborated by the attention score found in Figure 1 which was generated using full self-attention. Henaff et al. [17] show that orthogonal RNNs (orth-RNN) provide an optimal solution. We also consider expRNN [7] which does optimization in the unitary space and is so far the best purely performing recurrent model for large time delays for this task.

Table 5 (Appendix D) reports test performances of orth-RNN, expRNN, MemRNN, SAB, RelRNN and RelLSTM for $T = \{100, 200, 300, 500, 1000, 2000\}$ on the Copy Task. We find that orth-RNN solves this task up to $T = 500$, but that accuracy decays beyond that point, similarly to LSTM. RelRNN, RelLSTM, SAB and expRNN perfectly solve this task with **100**% accuracy for all $T$, while Fig 4 in Appendix D shows that RelRNN learn copy and denoise tasks with significantly fewer number of updates as compared to other baselines. MemRNN solves this task until $T = 100$ but overflows memory (OOM) afterwards.

**Transfer Copy task**: An important advantage of sparse attentive recurrent models such as RelRNN is that of generalization. This is illustrated by the Transfer Copy scores [19] where models are trained on Copy task for $T = 100$ and evaluated for $T > 100$. Table 1 shows results for the models listed above, in addition to h-detach [3], an LSTM-based model with improved gradient propagation. Importantly, where purely recurrent networks performed well on the original task, all fail to transfer, with discrepancy growing with $T$. As expected, MemRNN and SAB keep good performance but RelRNN outperforms them, with almost perfect performance for all $T$. While both SAB and RelRNN use sparse memory storage and retrieval, the distinguishing factor is RelRNN's use of relevancy screening, indicating it's importance for transfer. The performance of RelLSTM on Transfer Copy is exactly the same as RelRNN.

**Denoise task** Jing et al. [21]: This generalizes the Copy task as the symbols that need to be copied are now randomly distributed among the $T$ time steps, requiring the model to selectively pick the inputs that need to be copied. We test our method against all the previously mentioned models in addition to GORU [21] for various values of $T$ (Table 2). RelLSTM performs exactly as RelRNN and again, we see RelRNN maintain complete performance across all $T$ values, outperforming all purely recurrent models. MemRNN performs as RelRNN/RelLSTM but fails to train due to memory overflow beyong $T = 500$.

## 5.2 Tasks with dense temporal dependencies

In contrast to sparse information found in the tasks above, we now illustrate RelRNN and RelLSTM's performance on tasks with densely distributed information on long sequences.

Table 3: PTB and pMNIST results.

| Model | PTB Task | | pMNIST |
| | BPC | Accuracy | Accuracy |
| --- | --- | --- | --- |
| RNN | 1.56 | 66% | 90.4% |
| orth-RNN | 1.53 | 66% | 93.4% |
| expRNN | 1.49 | 68% | **96.6**% |
| RelRNN | **1.43** | **69**% | 92.8% |
| LSTM | **1.36** | **73**% | 91.1% |
| h-detach | - | - | 92.3% |
| SAB | 1.37 | - | 94.2% |
| RelLSTM | **1.36** | **73**% | **94.3**% |

Here, we perform tests on pMNIST [24], a variant of MNIST [25] where pixels are fed sequentially in a permuted order to the network, as well as character level Penn Tree Bank corpus (PTB) [27] where the next letter in a text needs to be predicted.

See Table 3 for results. Implementation details and further test data found in Appendix D, including attention heatmaps such as the ones found in Figure 1, showing dense attention for RelRNN in both tasks. We note that gated RNNs such as LSTMs are known to perform well here, and that orthogonal RNNs such as those tested here are also very good. The full attention model (MemRNN) fails to train on the optimization setup used here for both tasks, again due to overflow in memory.

## 6 Analysis

In this section we analyze the maximal GPU usage and gradient norm of $\|\nabla_{h_t} L\|$ across time $t$ for the Denoise Task. All the models were run using a NVIDIA TitanXP GPU and their peak usage was recorded in order to quantify the amount of computational resources used for each of them. We varied sequence length $T$ from 200 to 2000 in order to measure the trend in the usage. To measure propagating gradients as a function of $t$, we trained models on $T = 1000$ and computed $\log \|\nabla_{h_t} L\|$.

As illustrated in Figure 2 (center), we confirm MemRNN scales quadratically with $T$, same as SAB which shows an improvement but only by a constant factor. We also confirm that RelLSTM scales linearly with $T$ similar to RNN and LSTM. Figure 2 (left) shows that the gradient norms for RNN explode and for LSTM vanish as $t$ increases. The gradient norms of all attention models were stable, as expected from the results of Section 3. To better visualize the interplay between gradient norm and GPU usage, Figure 2 (right) shows the final averaged log gradient norm against Max GPU usage for different times $T = \{400, 600, 800\}$. As expected, purely recurrent models (RNN, LSTM) show very little GPU usage differences across distinct $T$ values, while their performance and gradients degrade with increasing $t$. Note that the RNN's gradients explode while the LSTM's vanish, both exponentially in $t$. Standard self attentive models (MemRNN, SAB) on the other hand, show opposite trends, with stable gradients but GPU usage quadratically increasing in $T$. As expected from Theorem 2 (Section 3), RelLSTM shows both stable gradients and stable GPU usage[2].

The optimal trade-off between memory usage and good gradient propagation achieved by RelLSTM highlights the importance of a dynamic memory that attempts to predict relevancy in order to only store exactly those events that help with learning. We note the Denoise task has a small number of relevant events and that not all tasks share this structure. Nevertheless, this experiment highlights how important resource gains can be made by shifting efforts from offsetting memory growth by a constant factor, to a relevancy screening method.

## 7 Conclusion & Discussion

Our main contribution is a formal analysis of gradient propagation in self-attention RNNs, from which we derive two quantities that are governing gradient propagation: sparsity and dependency depth. Meanwhile we identify event relevancy as a key concept to efficiently scale attentive systems to very long sequential computations. This is illustrated via a *Relevancy Screening Mechanism*, inspired by the cognitive process of memory consolidation in the brain, that efficiently selects network states, called relevant events, to be committed to long-term memory based on a screening heuristic operating on a fixed-size short-term memory buffer. We showcase the benefits of this mechanism in an attentive RNN and LSTM which we call RelRNN and RelLSTM respectively, using simple but illustrative numerical experiments, and demonstrate the optimal trade-off between memory usage and good gradient propagation it achieves.

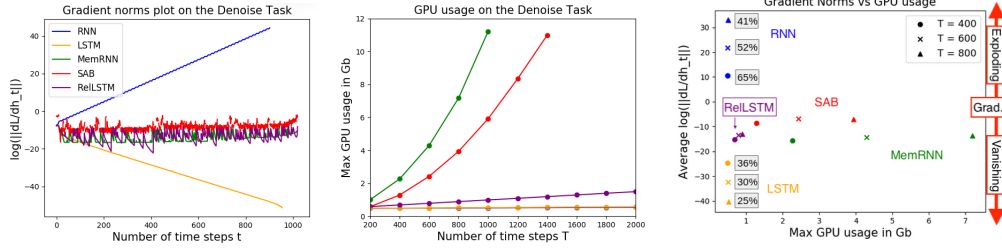

Figure 2: **(Left)** gradient norm plots of $\|\nabla_{h_t} L\|$ in log scale after training for Denoise Task with $t$ ranging from 0 (latest time step) to 1000 (furthest time step).**(Center)** Maximal GPU usage as a function of total sequence length $T$.**(Right)** Mean log gradient norm v.s. Max GPU usage for $T = 400, 600, 800$. Model testing accuracy is $100\%$ unless indicated by marker label (see Table 2).

As outlined in Sections 3 and 4, this trade-off is a reflection of the task-specific balance between sparsity and dependency depth parameters. While our proposed relevancy screening mechanism exploits "local" attention scores (measured while events are in short-term memory buffer), we acknowledge other types of relevancy could be formulated with heuristics better suited to distinct environments. For instance, promising directions include leveraging predictive coding principles to select "surprising events", or neural networks could be used to learn the screening function $C(i)$ in an end-to-end fashion.

## Broader Impact

We provide a framework for researchers to shape gradient propagation and memory footprint in self-attentive RNNs, which is helpful in tasks requiring ongoing online predictions that cannot be based on future inputs (i.e. in an online sequential setting) and where long-term credit assignment is crucial, such as various RL tasks [16, 20]. The added resource gains can save GPU hours and thus have a positive environmental impact. Along this line, we firmly believe that researchers should take environmental impact of model training seriously, and we are hopeful that our work contributes to this direction.

Meanwhile, the theoretical tools provided in the proofs lay the ground for more theoretical work on attentive systems to emerge in the future. More effective RNN models can amplify already existing biases in RNN-based NLP systems through an increased exposure to bias. Finally, we cannot exclude that the cognitive inductive bias we use to build our relevancy screening mechanism may induce prediction quality disparity (e.g. in language modelling) because of the memory tokens it throws away.

## Acknowledgments and Disclosure of Funding

We would like to thank Gauthier Gidel, Naz Sepah, Yassine Yaakoubi, Sarthak Mittal, and Chen Sun for useful discussions. BK acknowledges IVADO Masters Excellence Scholarship. YB acknowledges support from CIFAR, Microsoft and NSERC. GL is funded by an NSERC Discovery Grant (RGPIN-2018-04821), an FRQNT Young Investigator Startup Program (2019-NC-253251), and an FRQS Research Scholar Award, Junior 1 (LAJGU0401-253188).

## Footnotes

[2]The measurements for both GPU usage and gradient norm are identical for both RelLSTM and RelRNN.

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
