[Supplementary Material · attention_NeurIPS_2020_paper-13-45.pdf]



# A   Theoretical analysis of gradient propagation

## A.1   Notational convention

In this paper, we use the notation $\frac{df}{dx}$ to denote the total derivative of $f$ with respect to $x$, and $\frac{\partial f}{\partial x}$ to denote the partial derivative of $f$ with respect to $x$.

If we assume $f : \mathbb{R}^n \to \mathbb{R}^m$, and $x \in \mathbb{R}^n$, then $\frac{df}{dx}$ denotes the Jacobian matrix $J_f$ such that

$$(J_f)_{ij} = \frac{df_i}{dx_j} \tag{6}$$

In particular, with this notation, we have that if a function $L : \mathbb{R}^m \to \mathbb{R}$, and $y \in \mathbb{R}^m$ then $\frac{dL}{dy}$ is a row vector, while the conventional notation for $\nabla_y L$ indicates a column vector. In other words, $(\nabla_y L)^T = \frac{dL}{dy}$. Hence if $L$ is a function of $f(x)$, then

$$\frac{dL}{dx} = \frac{dL}{df} \cdot \frac{df}{dx} \tag{7}$$

while

$$\nabla_x L = \left(\frac{df}{dx}\right)^T \cdot \nabla_{f(x)} L = J_f^T \cdot \nabla_{f(x)} L \tag{8}$$

Similarly, we have that $\frac{\partial L}{\partial y}$ is a row vector.

## A.2   Preliminary results

Let

$$s_t = \psi_t(h_1, h_2, \ldots, h_t, s_{t-1}) \tag{9}$$

where

$$h_{i+1} = \phi(V s_i + U x_{i+1} + b) \tag{10}$$

**Lemma 1.** *For all $t, k \geq 0$, we have*

$$\frac{ds_{t+k+1}}{dh_t} = \frac{\partial s_{t+k+1}}{\partial h_t} + \left(\sum_{j=0}^{k} \frac{\partial s_{t+k+1}}{\partial h_{t+j+1}} \frac{dh_{t+j+1}}{dh_t}\right) + \frac{\partial s_{t+k+1}}{\partial s_{t+k}} \frac{ds_{t+k}}{dh_t} \tag{11}$$

*Proof.* Follows directly from the following multivariable chain rule: if

$$g(t) = f(g_1(t), g_2(t), \ldots, g_n(t)) \tag{12}$$

then

$$\frac{dg}{dt} = \sum_{i=1}^{n} \frac{\partial f}{\partial g_i} \frac{dg_i}{dt} \tag{13}$$

$\square$

**Lemma 2.** *If we further denote the Jacobian matrix $J_k = \frac{\partial s_{k+1}}{\partial h_k}$, then we get that for all $t, k \geq 0$, we have*

$$\frac{ds_{t+k+1}}{dh_t} = \frac{\partial s_{t+k+1}}{\partial h_t} + \sum_{j=0}^{k} \left(\frac{\partial s_{t+k+1}}{\partial h_{t+j+1}} \cdot J_{t+j} + 1_{j=k} \cdot \frac{\partial s_{t+k+1}}{\partial s_{t+k}}\right) \cdot \frac{ds_{t+j}}{dh_t} \tag{14}$$

*Proof.* Follows directly from the observation that

$$\frac{dh_{t+j+1}}{dh_t} = \frac{\partial h_{t+j+1}}{\partial s_{t+j}} \frac{ds_{t+j}}{dh_t} = J_{t+j} \cdot \frac{ds_{t+j}}{dh_t} \tag{15}$$

$\square$

**Remark 1.** *Let us denote*

$$C_{k+1}^{(t)} = \frac{ds_{t+k+1}}{dh_t} \tag{16}$$

$$E_{k+1}^{(t)} = \frac{\partial s_{t+k+1}}{\partial h_t} \tag{17}$$

*and*

$$F_{k+1,j}^{(t)} = \frac{\partial s_{t+k+1}}{\partial h_{t+j+1}} \cdot J_{t+j} + 1_{j=k} \cdot \frac{\partial s_{t+k+1}}{\partial s_{t+k}} \tag{18}$$

*and thus the recursion formula in Lemma 2 rewrites as*

$$C_{k+1}^{(t)} = E_{k+1}^{(t)} + \sum_{j=0}^{k} F_{k+1,j}^{(t)} \cdot C_j^{(t)} \tag{19}$$

*The next two results highlight how to solve this recursion.*

**Lemma 3.** *Let* $C_i, E_i, F_{i,j} \in \mathbb{R}^{n \times n}$ *such that for all* $k \geq 0$*, we have*

$$C_{k+1} = E_{k+1} + \sum_{j=0}^{k} F_{k+1,j} \cdot C_j \tag{20}$$

*Then for all* $k \geq 1$*, we have*

$$\boxed{C_k = \xi_{0:k} C_0 + \sum_{r=1}^{k} \xi_{r:k} E_r} \tag{21}$$

*where*

$$\xi_{r:k} = \sum_{s=1}^{k-r} \xi_{r:k}(s) \tag{22}$$

*with*

$$\xi_{r:k}(s) = \sum_{r=i_1<...<i_{s+1}=k} F_{i_{s+1},i_s} \cdot F_{i_{s-1},i_{s-2}} \cdot ... \cdot F_{i_2,i_1} \tag{23}$$

*and* $\xi_{k:k} = Id$*.*

*Proof.* Let us prove the statement by induction on $k \geq 1$.
For $k = 1$, we have

$$C_1 = E_1 + F_{1,0} C_0 = \xi_{1:1} E_1 + \xi_{0:1} C_0 \tag{24}$$

Now let us assume the statement to be true for $k$, then we get

$$C_{k+1} = E_{k+1} + \sum_{j=0}^{k} F_{k+1,j} \cdot \left( \xi_{0:j} C_0 + \sum_{r=1}^{j} \xi_{r:j} E_r \right) \tag{25}$$

$$= E_{k+1} + \left( \sum_{j=0}^{k} F_{k+1,j} \cdot \xi_{0:j} \right) \cdot C_0 + \sum_{j=0}^{k} \sum_{r=1}^{j} F_{k+1,j} \xi_{r:j} E_r \tag{26}$$

$$= E_{k+1} + \xi_{0:k+1} C_0 + \sum_{r=1}^{k} \left( \sum_{j=r}^{k} F_{k+1,j} \xi_{r:j} \right) \cdot E_r \tag{27}$$

$$= \xi_{k+1:k+1} E_{k+1} + \xi_{0:k+1} C_0 + \sum_{r=1}^{k} \xi_{r:k+1} E_r \tag{28}$$

$$= \xi_{0:k+1} C_0 + \sum_{r=1}^{k+1} \xi_{r:k+1} E_r \tag{29}$$

$$\tag{30}$$

$\square$

---

**Lemma 4.** *If we further assume that $C_0 = E_0$, then we have for all $k \geq 1$*

$$C_k = E_k + \sum_{s=1}^{k} \sum_{q=s}^{k} \xi_{k-q:k}(s) E_{k-q} \tag{31}$$

*Proof.* Using the previous lemma, we get

$$C_k = E_k + \sum_{s'=1}^{k} \xi_{0:k}(s') C_0 + \sum_{r=1}^{k-1} \sum_{s=1}^{k-r} \xi_{r:k}(s) E_r \tag{32}$$

Using the assumption $C_0 = E_0$, we get

$$C_k = E_k + \sum_{s'=1}^{k} \xi_{0:k}(s') E_0 + \sum_{r=1}^{k-1} \sum_{s=1}^{k-r} \xi_{r:k}(s) E_r \tag{33}$$

$$= E_k + \sum_{r=0}^{k-1} \sum_{s=1}^{k-r} \xi_{r:k}(s) E_r \tag{34}$$

$$\tag{35}$$

Now let us put $q = k - r$, we get

$$C_k = E_k + \sum_{q=1}^{k} \sum_{s=1}^{q} \xi_{k-q:k}(s) E_{k-q} \tag{36}$$

$$= E_k + \sum_{s=1}^{k} \sum_{q=s}^{k} \xi_{k-q:k}(s) E_{k-q} \tag{37}$$

$$\tag{38}$$

$\square$

---

**Remark 2.** *First, note that Lemma 4 applies here, since $C_0^{(t)} = E_0^{(t)}$, and thus*

$$C_k^{(t)} = E_k^{(t)} + \sum_{s=1}^{k} \sum_{q=s}^{k} \xi_{k-q:k}^{(t)}(s) E_{k-q}^{(t)} \tag{39}$$

*The idea of Lemma 4 was to regroup all terms with the same number of $F$ factors (where each $F$ contains a Jacobian matrix $J_k$ which contains the connectivity matrix $V$ of the recurrent net). One could roughly perceive the term*

$$\sum_{q=s}^{k} \xi_{k-q:k}^{(t)}(s) E_{k-q}^{(t)} \tag{40}$$

*as being the term of degree $s$ for $s = 1, 2, \ldots, k$ and $E_k^{(t)}$ the term of degree $0$. This will allow us to consider the terms $C$ roughly as a polynomial in $V$ and we can look the asymptotic behaviour of each of the coefficients of this polynomial individually. This will then give us a very good understanding on how the distribution of the attention weights are affecting the magnitude of total gradient.*

---

**Proposition 1.** *For all $t \geq 1$, and all $k \geq 0$, we have that*

$$\frac{ds_{t+k}}{dh_t} = \sum_{s=0}^{k} \bar{\xi}_{o:k}^{(t)}(s) \tag{41}$$

*where for all $s \geq 1$,*

$$\bar{\xi}_{o:k}^{(t)}(s) = \sum_{0 \leq i_1 < \ldots < i_s < k} F_{k,i_s}^{(t)} \cdot F_{i_s,i_{s-1}}^{(t)} \cdot \ldots \cdot F_{i_2,i_1}^{(t)} \cdot E_{i_1}^{(t)} \tag{42}$$

*and where $\bar{\xi}_{o:k}^{(t)}(0) = E_k^{(t)}$. With for all $k \geq 0$ we have*

$$E_k^{(t)} = \frac{\partial s_{t+k}}{\partial h_t} \tag{43}$$

*and for all $k \geq j$ we have*

$$F_{k+1,j}^{(t)} = \frac{\partial s_{t+k+1}}{\partial h_{t+j+1}} \cdot J_{t+j} + 1_{j=k} \cdot \frac{\partial s_{t+k+1}}{\partial s_{t+k}} \tag{44}$$

*Proof.* Let $t \geq 1$, and recall that we defined $C_k^{(t)} = \frac{ds_{t+k}}{dh_t}$, for all $k \geq 0$.

As already pointed out, we know that $C_0^{(t)} = E_0^{(t)}$ (thus the claim holds for $k = 0$).

Then by Lemma 4, we know that for all $k \geq 1$ we have

$$C_k^{(t)} = E_k^{(t)} + \sum_{s=1}^{k} \sum_{q=s}^{k} \xi_{k-q:k}^{(t)}(s) E_{k-q}^{(t)} \tag{45}$$

$$= \bar{\xi}_{o:k}^{(t)}(0) + \sum_{s=1}^{k} \sum_{q=s}^{k} \sum_{k-q=i_1 < \ldots < i_{s+1}=k} F_{k,i_s}^{(t)} \cdot F_{i_s,i_{s-1}}^{(t)} \cdot \ldots \cdot F_{i_2,i_1}^{(t)} \cdot E_{i_1}^{(t)} \tag{46}$$

$$= \bar{\xi}_{o:k}^{(t)}(0) + \sum_{s=1}^{k} \sum_{0 \leq i_1 < \ldots < i_s < k} F_{k,i_s}^{(t)} \cdot F_{i_s,i_{s-1}}^{(t)} \cdot \ldots \cdot F_{i_2,i_1}^{(t)} \cdot E_{i_1}^{(t)} \tag{47}$$

$$= \bar{\xi}_{o:k}^{(t)}(0) + \sum_{s=1}^{k} \bar{\xi}_{o:k}^{(t)}(s) \tag{48}$$

$$= \sum_{s=0}^{k} \bar{\xi}_{o:k}^{(t)}(s) \tag{49}$$

$\square$

**Remark 3.** *In what follows the main emphasis will be to calculate the $F_{i,j}^{(t)}$ and $E_i^{(t)}$ terms explicitly, since they are the building blocks of the mentioned polynomials in 2.*

*We will assume that*

$$s_t = f(h_t, c_t) \tag{50}$$

*with*

$$c_t = \alpha_{1,t}h_1 + \alpha_{2,t}h_2 + \ldots + \alpha_{t,t}h_t \tag{51}$$

*and*

$$\alpha_{j,t} = \frac{\exp(e_{j,t})}{\sum_{i=1}^{t} \exp(e_{i,t})} \tag{52}$$

*where*

$$e_{i,t} = a(s_{t-1}, h_i) \tag{53}$$

*Let us recall that for all $k \geq 0$ we have*

$$E_k^{(t)} = \frac{\partial s_{t+k}}{\partial h_t} \tag{54}$$

*and for all $k \geq j$ we have*

$$F_{k+1,j}^{(t)} = \frac{\partial s_{t+k+1}}{\partial h_{t+j+1}} \cdot J_{t+j} + 1_{j=k} \cdot \frac{\partial s_{t+k+1}}{\partial s_{t+k}} \tag{55}$$

---

**Lemma 5.** *With the assumption of Remark 3, we have that for all $t \geq 2$*

$$\frac{\partial s_t}{\partial s_{t-1}} = \partial_2 f(h_t, c_t) \cdot \left( \sum_{i=1}^{t} \alpha_{i,t} Y_{i,t} \right) \tag{56}$$

*where $\partial_2 f$ is the the partial derivative of $f$ with respect to the second variable, and where we define*

$$Y_{i,t} = h_i \cdot \left( \frac{\partial e_{i,t}}{\partial s_{t-1}} - \sum_{j=1}^{t} \alpha_{j,t} \cdot \frac{\partial e_{j,t}}{\partial s_{t-1}} \right) \tag{57}$$

*Proof.*

$$\frac{\partial s_t}{\partial s_{t-1}} = \partial_2 f(h_t, c_t) \cdot \frac{\partial c_t}{\partial s_{t-1}} \tag{58}$$

$$= \partial_2 f(h_t, c_t) \cdot \left[ \sum_{i=1}^{t} h_i \cdot \left( \frac{\partial \alpha_{i,t}}{\partial s_{t-1}} \right) \right] \tag{59}$$

$$= \partial_2 f(h_t, c_t) \cdot \left[ \sum_{i=1}^{t} h_i \cdot \left( \sum_{j=1}^{t} \frac{\partial \alpha_{i,t}}{\partial e_{j,t}} \cdot \frac{\partial e_{j,t}}{\partial s_{t-1}} \right) \right] \tag{60}$$

$$= \partial_2 f(h_t, c_t) \cdot \left[ \sum_{i=1}^{t} h_i \cdot \left( \sum_{j=1}^{t} \alpha_{i,t}(1_{i=j} - \alpha_{j,t}) \cdot \frac{\partial e_{j,t}}{\partial s_{t-1}} \right) \right] \tag{61}$$

$$= \partial_2 f(h_t, c_t) \cdot \left[ \sum_{i=1}^{t} \alpha_{i,t} h_i \left( \frac{\partial e_{i,t}}{\partial s_{t-1}} - \sum_{j=1}^{t} \alpha_{j,t} \frac{\partial e_{j,t}}{\partial s_{t-1}} \right) \right] \tag{62}$$

$$= \partial_2 f(h_t, c_t) \cdot \left( \sum_{i=1}^{t} \alpha_{i,t} Y_{i,t} \right) \tag{63}$$

$\square$

**Lemma 6.** *With the assumption of Remark 3, we have that for all $k \geq j$:*

$$\frac{\partial s_k}{\partial h_j} = 1_{k=j} \cdot \partial_1 f(h_k, c_k) + \alpha_{j,k} \partial_2 f(h_k, c_k) \cdot (I + X_{j,k}) \tag{64}$$

*where $\partial_1 f$ and $\partial_2 f$ are the partial derivatives of $f$ with respect to the first and second variable, respectively, and where we define*

$$X_{j,k} = \left( h_j - \sum_{i=1}^{k} h_i \alpha_{i,k} \right) \cdot \frac{\partial e_{j,k}}{\partial h_j} \tag{65}$$

*Proof.*

$$\frac{\partial s_k}{\partial h_j} = 1_{k=j} \cdot \partial_1 f(h_k, c_k) \cdot \frac{\partial h_k}{\partial h_k} + \partial_2 f(h_k, c_k) \cdot \frac{\partial c_k}{\partial h_j} \tag{66}$$

$$= 1_{k=j} \cdot \partial_1 f(h_k, c_k) + \partial_2 f(h_k, c_k) \cdot \left[ \alpha_{j,k} \cdot I + \sum_{i=1}^{k} h_i \cdot \frac{\partial \alpha_{i,k}}{\partial h_j} \right] \tag{67}$$

$$= 1_{k=j} \cdot \partial_1 f(h_k, c_k) + \partial_2 f(h_k, c_k) \cdot \left[ \alpha_{j,k} \cdot I + \sum_{i=1}^{k} h_i \cdot \frac{\partial \alpha_{i,k}}{\partial e_{j,k}} \frac{\partial e_{j,k}}{\partial h_j} \right] \tag{68}$$

$$= 1_{k=j} \cdot \partial_1 f(h_k, c_k) + \partial_2 f(h_k, c_k) \cdot \left[ \alpha_{j,k} \cdot I + \sum_{i=1}^{k} h_i \cdot \alpha_{i,k}(1_{i=j} - \alpha_{j,k}) \frac{\partial e_{j,k}}{\partial h_j} \right] \tag{69}$$

$$= 1_{k=j} \cdot \partial_1 f(h_k, c_k) + \partial_2 f(h_k, c_k) \cdot \left[ \alpha_{j,k} \cdot I + \left( h_j \alpha_{j,k} - \alpha_{j,k} \sum_{i=1}^{k} h_i \cdot \alpha_{i,k} \right) \frac{\partial e_{j,k}}{\partial h_j} \right] \tag{70}$$

$$= 1_{k=j} \cdot \partial_1 f(h_k, c_k) + \alpha_{j,k} \partial_2 f(h_k, c_k) \cdot \left[ I + \left( h_j - \sum_{i=1}^{k} h_i \cdot \alpha_{i,k} \right) \frac{\partial e_{j,k}}{\partial h_j} \right] \tag{71}$$

$$= 1_{k=j} \cdot \partial_1 f(h_k, c_k) + \alpha_{j,k} \partial_2 f(h_k, c_k) \cdot (I + X_{j,k}) \tag{72}$$

$\square$

**Corollary 1.** *With the assumption of Remark 3, and the notations of lemma 5 and 6, we have for all $k' \geq 0$,*

$$E_{k'}^{(t)} = 1_{k'=0} \partial_1 f(h_t, c_t) + \alpha_{t,t+k'} \partial_2 f(h_{t+k'}, c_{t+k'}) \cdot [I + X_{t,t+k'}] \tag{73}$$

*and for all $k \geq j$,*

$$F_{k+1,j}^{(t)} = \alpha_{t+j+1,t+k+1} \cdot \partial_2 f(h_{t+k+1}, c_{t+k+1}) \cdot [I + X_{t+j+1,t+k+1}] \cdot J_{t+j} \tag{74}$$

$$+ 1_{k=j} \cdot \left( \partial_1 f(h_{t+k+1}, c_{t+k+1}) J_{t+j} + \partial_2 f(h_{t+k+1}, c_{t+k+1}) \cdot \left[ \sum_{i=1}^{t+k+1} \alpha_{i,t+k+1} Y_{i,t+k+1} \right] \right) \tag{75}$$

*Proof.* Applying lemma 6, we get that for all $k \geq 0$,

$$E_{k'}^{(t)} = \frac{\partial s_{t+k'}}{\partial h_t} \tag{76}$$

$$= 1_{k'=0} \cdot \partial_1 f(h_t, c_t) + \alpha_{t,t+k'} \cdot \partial_2 f(h_{t+k'}, c_{t+k'}) \cdot [I + X_{t,t+k'}] \tag{77}$$

$$\tag{78}$$

and then by applying lemma 5 and 6, we get that for all $k \geq j$,

$$F_{k+1,j}^{(t)} = \frac{\partial s_{t+k+1}}{\partial h_{t+j+1}} \cdot J_{t+j} + 1_{j=k} \cdot \frac{\partial s_{t+k+1}}{\partial s_{t+k}} \tag{79}$$

$$= [1_{k=j} \partial_1 f(h_{t+k+1}, c_{t+k+1}) \tag{80}$$

$$+ \alpha_{t+j+1,t+k+1} \partial_2 f(h_{t+k+1}, c_{t+k+1}) \cdot (I + X_{t+j+1,t+k+1})] \cdot J_{t+j} \tag{81}$$

$$+ 1_{k=j} \cdot \partial_2 f(h_{t+k+1}, c_{t+k+1}) \cdot \left( \sum_{i=1}^{t+k+1} \alpha_{i,t+k+1} Y_{i,t+k+1} \right) \tag{82}$$

$$= \alpha_{t+j+1,t+k+1} \cdot \partial_2 f(h_{t+k+1}, c_{t+k+1}) \cdot [I + X_{t+j+1,t+k+1}] \cdot J_{t+j} \tag{83}$$

$$+ 1_{k=j} \cdot \left( \partial_1 f(h_{t+j+1}, c_{t+k+1}) J_{t+j} + \partial_2 f(h_{t+k+1}, c_{t+k+1}) \cdot \left[ \sum_{i=1}^{t+k+1} \alpha_{i,t+k+1} Y_{i,t+k+1} \right] \right) \tag{84}$$

$$\square$$

---

**Proposition 2.** *We can rewrite for all $k' \geq 0$ and all $k \geq j \geq 0$*

$$E_{k'}^{(t)} = \alpha_{t,t+k'} \cdot \tilde{D}_{k',0}^{(t)} + 1_{k'=0} \tilde{R}_0^{(t)} \tag{85}$$

$$F_{k+1,j}^{(t)} = \alpha_{t+j+1,t+k+1} \cdot D_{k+1,j}^{(t)} + 1_{k=j} \cdot R_{k+1}^{(t)} \tag{86}$$

*where*

$$D_{k+1,j+1}^{(t)} = \partial_2 f(h_{t+k+1}, c_{t+k+1}) \cdot [I + X_{t+j+1,t+k+1}] \cdot J_{t+j} \tag{87}$$

$$R_{k+1}^{(t)} = \partial_1 f(h_{t+k+1}, c_{t+k+1}) \cdot J_{t+k} + \partial_2 f(h_{t+k+1}, c_{t+k+1}) \cdot \left[ \sum_{i=1}^{t+k+1} \alpha_{i,t+k+1} Y_{i,t+k+1} \right] \tag{88}$$

$$\tilde{D}_{k'}^{(t)} = \partial_2 f(h_{t+k'}, c_{t+k'}) \cdot [I + X_{t,t+k'}] \tag{89}$$

$$\tilde{R}_0^{(t)} = \partial_1 f(h_t, c_t) \tag{90}$$

*while $X_{i,i'}$ and $Y_{i,i'}$ are defined as in lemma 5 and 6.*

*Proof.* Follows straight from Corollary 1. $\square$

---

**Remark 4.** *If we are further assuming that*

$$s_t = f(h_t, c_t) = h_t + c_t \tag{91}$$

*then for all $k \geq 0$, we have*

$$E_k^{(t)} = 1_{k=0} \cdot I + \alpha_{t,t+k} \cdot [I + X_{t,t+k}] \tag{92}$$

*and for all $k \geq j$, we have*

$$F_{k+1,j}^{(t)} = \alpha_{t+j+1,t+k+1} \cdot [I + X_{t+j+1,t+k+1}] \cdot J_{t+j} + 1_{k=j} \cdot \left( J_{t+j} + \left[ \sum_{i=1}^{t+k+1} \alpha_{i,t+k+1} Y_{i,t+k+1} \right] \right) \tag{93}$$

*Proof.* This follows directly form corollary 1 and the observation that

$$\partial_1 f(h_t, c_t) = \partial_2 f(h_t, c_t) = I \tag{94}$$

$$\square$$

**Remark 5.** *If we are further assuming that*

$$e_{j,t} = a(s_{t-1}, h_j) = v_a^T \cdot \tanh\left(W_a s_{t-1} + U_a h_j\right) \tag{95}$$

*as done in [4], we get that*

$$\frac{\partial e_{j,t}}{\partial h_j} = v_a^T \cdot diag[1 - \tanh^2\left(W_a s_{t-1} + U_a h_j\right)] \cdot U_a \tag{96}$$

*and*

$$\frac{\partial e_{j,t}}{\partial s_{t-1}} = v_a^T \cdot diag[1 - \tanh^2\left(W_a s_{t-1} + U_a h_j\right)] \cdot W_a \tag{97}$$

*which we can plug into the definitions of $X_{j,k}$ and $Y_{j,k}$ to get explicit expressions for matrices $E_{k'}^{(t)}$ and $F_{k+1,j}^{(t)}$.*

---

**Lemma 7.** *If, with the assumptions Remark 3, we assume that for all $i, t \geq 1$, we have $e_{i,t} = a(s_{t-1}, h_i, \theta)$ depending on some parameter $\theta \in \mathbb{R}^{N \times M}$, then we have*

$$\frac{dL}{d\theta} = \sum_{j,t} \alpha_{j,t} \cdot \frac{dL}{ds_t} \cdot \partial_2 f(h_t, c_t) \cdot h_j \cdot \left[\sum_i (1_{i=j} - \alpha_{i,t}) \cdot \frac{\partial e_{i,t}}{\partial \theta}\right] \tag{98}$$

*Proof.* If we denote $\theta^{(i,t)}$ to be the parameter for $e_{i,t}$, then we have

$$\frac{dL}{d\theta} = \sum_{i,t} \frac{dL}{d\theta^{(i,t)}} \tag{99}$$

$$= \sum_{i,j,t} \frac{dL}{d\alpha_{j,t}} \cdot \frac{\partial \alpha_{j,t}}{\partial e_{i,t}} \cdot \frac{\partial e_{i,t}}{\partial \theta^{(i,t)}} \tag{100}$$

$$= \sum_{i,j,t} \alpha_{j,t}(1_{i=j} - \alpha_{i,t}) \cdot \frac{dL}{d\alpha_{j,t}} \cdot \frac{\partial e_{i,t}}{\partial \theta} \tag{101}$$

where

$$\frac{dL}{d\alpha_{j,t}} = \frac{dL}{ds_t} \cdot \frac{\partial s_t}{\partial c_t} \cdot \frac{\partial c_t}{\partial \alpha_{j,t}} = \frac{dL}{ds_t} \cdot \partial_2 f(h_t, c_t) \cdot h_j \tag{102}$$

Hence

$$\frac{dL}{d\theta} = \sum_{i,j,t} \alpha_{j,t}(1_{i=j} - \alpha_{i,t}) \cdot \frac{dL}{ds_t} \cdot \partial_2 f(h_t, c_t) \cdot h_j \cdot \frac{\partial e_{i,t}}{\partial \theta} \tag{103}$$

$$= \sum_{j,t} \alpha_{j,t} \cdot \frac{dL}{ds_t} \cdot \partial_2 f(h_t, c_t) \cdot h_j \cdot \left[\sum_i (1_{i=j} - \alpha_{i,t}) \cdot \frac{\partial e_{i,t}}{\partial \theta}\right] \tag{104}$$

$\square$

---

**Lemma 8.** *Let us recall that for all $t \geq 0$, we have*

$$h_{t+1} = \phi(\underbrace{V s_t + U x_{t+1} + b}_{=a_t}) \tag{105}$$

*where $\phi$ is a non-linear activation function, $V \in \mathbb{R}^{n \times n}$, $U \in \mathbb{R}^{n \times m}$ and $b \in \mathbb{R}^n$. Then we have that*

$$\left[\frac{dL}{dV}, \frac{dL}{dU}, \frac{dL}{db}\right] = \sum_{t=1}^{T} [s_{t-1}, x_t, 1] \cdot \frac{dL}{dh_t} \cdot diag(\phi'(a_t)) \tag{106}$$

*Proof.* Let us denote $V^{(t)}, U^{(t)}, b^{(t)}$ the matrices $V, U, b$ of $a_{t-1}$ respectively, then

$$\left[\frac{dL}{dV}, \frac{dL}{dU}, \frac{dL}{db}\right] = \sum_t \left[\frac{dL}{dV^{(t)}}, \frac{dL}{dU^{(t)}}, \frac{dL}{db^{(t)}}\right] \tag{107}$$

$$= \sum_t [s_{t-1}, x_t, 1] \cdot \frac{dL}{da_{t-1}} \tag{108}$$

$$= \sum_t [s_{t-1}, x_t, 1] \cdot \frac{dL}{dh_t} \cdot \frac{dh_t}{da_{t-1}} \tag{109}$$

$$= \sum_t [s_{t-1}, x_t, 1] \cdot \frac{dL}{dh_t} \cdot \text{diag}(\phi'(a_{t-1})) \tag{110}$$

$\square$

---

**Remark 6.** *Combining the fact that $\frac{dL}{dh_t} = \frac{dL}{ds_T}\frac{ds_T}{dh_t}$, the results from propositions 1 and 2, with lemma 8, we see that attention weights $\alpha_{i,t}$ which are very close to 0, do not contribute to the gradient and the learning of $V, U$ and $b$.*

*Similarly, it follows directly from lemma 7, that attention weights $\alpha_{i,t}$ which are very close to 0, do not contribute to the gradient and the learning of any parameters $\theta$ of the alignment function $e_{i,t} = a(s_{t-1}, h_i, \theta)$. In case we have an alignment function as in remark 5, these parameters are $W_a, U_a$ and $v_a$.*

*If we have the case where one state $h_i$ is such that all attention weights $\alpha_{i,t} \approx 0$ for all $t \geq i$, then we can see that $h_i$ does not contribute to the gradient and learning to any parameters be it parameters from the recurrence or the alignment function.*

*In practice we have observed that in the majority of tasks, most states $h_i$ fall in either of two categories:*

- *$\alpha_{i,t}$ is sufficiently bounded away from 0 for most $t \geq i$, and thus contributes to learning. This is what we call a "relevant state".*

- *$\alpha_{i,t} \approx 0$ for almost all $t \geq i$, and thus doesn't contribute much to learning, and the gradient can be approximated by assuming $\alpha_{i,t} = 0$ for all $t \geq i$. This is what he call a "non-relevant state".*

*This observation is what lead us to the intuition that we can approximate the gradient, by decomposing it via proposition 1, into gradient paths involving only skip connections between "relevant states".*

---

## A.3 Uniform attention case

**Remark 7.** *In this subsection, we are going to assume:*

- *no non-linearity in the hidden-to-hidden connection: $J_t = V$ for all $t$.*

- *all assumptions from Remark 3.*

- *uniform attention: $\alpha_{i,t} = 1/t$ for all $t \geq 1$.*

---

### A.3.1 Overview

**Remark 8.** *Recalling corollary 1, together the main proposition 1 form last section, we can hope to simplify these expressions using the new assumptions from the previous remark 7. Recalling*

*expression from lemma 5 and 6:*

$$X_{j,t} = \left( h_j - \sum_{i=1}^{t} h_i \alpha_{i,t} \right) \cdot \frac{\partial e_{j,t}}{\partial h_j} \tag{111}$$

$$= \left( h_j - \frac{1}{t} \sum_{i=1}^{t} h_i \right) \cdot \frac{\partial e_{j,t}}{\partial h_j} \tag{112}$$

*Hence, for our calculations we are going to assume that $\left( h_j - \frac{1}{t} \sum_{i=1}^{t} h_i \right) \approx 0$, and thus $X_{j,t} \approx 0$ for all $1 \leq j \leq t$. Similarly,*

$$\sum_{i=1}^{t} \alpha_{i,t} Y_{i,t} = \sum_{i=1}^{t} \alpha_{i,t} h_i \cdot \left( \frac{\partial e_{i,t}}{\partial s_{t-1}} - \sum_{j=1}^{t} \alpha_{j,t} \cdot \frac{\partial e_{j,t}}{\partial s_{t-1}} \right) \tag{113}$$

$$= \frac{1}{t} \sum_{i=1}^{t} h_i \cdot \left( \frac{\partial e_{i,t}}{\partial s_{t-1}} - \sum_{j=1}^{t} \frac{1}{t} \cdot \frac{\partial e_{j,t}}{\partial s_{t-1}} \right) \tag{114}$$

$$= \frac{1}{t} \sum_{i=1}^{t} h_i \cdot \frac{\partial e_{i,t}}{\partial s_{t-1}} - \frac{1}{t} \sum_{j=1}^{t} \left( \frac{1}{t} \sum_{i=1}^{t} h_i \right) \cdot \frac{\partial e_{j,t}}{\partial s_{t-1}} \tag{115}$$

$$= \frac{1}{t} \sum_{i=1}^{t} h_i \cdot \frac{\partial e_{i,t}}{\partial s_{t-1}} - \frac{1}{t} \sum_{i=1}^{t} \left( \frac{1}{t} \sum_{j=1}^{t} h_j \right) \cdot \frac{\partial e_{i,t}}{\partial s_{t-1}} \tag{116}$$

$$= \frac{1}{t} \sum_{i=1}^{t} \left( h_i - \frac{1}{t} \sum_{j=1}^{t} h_j \right) \cdot \frac{\partial e_{i,t}}{\partial s_{t-1}} \tag{117}$$

$$\approx 0 \tag{118}$$

*Recalling the expression from corollary 1 and that $f(h_t, c_t) = h_t + c_t$ by remark 3, and that $J_t = V$ for all $t$, this will give for all $k' \geq 0$*

$$E_{k'}^{(t)} = \left( \frac{1}{t + k'} + 1_{k'=0} \right) \cdot I \tag{119}$$

*and for all $k \geq j$, we get*

$$F_{k+1,j}^{(t)} = \left( \frac{1}{t + k + 1} + 1_{k=j} \right) \cdot V \tag{120}$$

*Hence by recalling proposition 1, the main expression of interest becomes*

$$\frac{ds_{t+k}}{dh_t} = \sum_{s=0}^{k} \bar{\xi}_{0:k}^{(t)}(s) = \sum_{s=0}^{k} V^s \cdot \chi_{0:k}^{(t)}(s) \tag{121}$$

*where*

$$\chi_{0:k}^{(t)}(s) \overset{\text{def}}{=} \sum_{0 \leq i_1 < \ldots < i_s < k} \left( \frac{1}{t+k} + 1_{k-i_s=1} \right) \cdot \left( \frac{1}{t+i_s} + 1_{i_s-i_{s-1}=1} \right) \cdot \ldots \tag{122}$$

$$\ldots \cdot \left( \frac{1}{t+i_2} + 1_{i_2-i_1=1} \right) \cdot \left( \frac{1}{t+i_1} + 1_{i_1=0} \right) \tag{123}$$

---

**Remark 9.** *The goal is thus to have a good estimation of the terms*

$$\chi_{0:k}^{(t)}(s) \tag{124}$$

*in order to then find an asymptotic estimation for*

$$\frac{ds_{t+k}}{dh_t} = \sum_{s=0}^{k} V^s \cdot \chi_{0:k}^{(t)}(s) \tag{125}$$

*as $k \to \infty$. In order to do so, we will adopt the following strategy:*

**Step 1.** *Estimate the expression*

$$\omega_{l:k}^{(t)}(s) \overset{\text{def}}{=} \sum_{l \le i_1 < \ldots < i_s < k} \frac{1}{t + i_s} \cdot \frac{1}{t + i_{s-1}} \cdot \ldots \cdot \frac{1}{t + i_2} \cdot \frac{1}{t + i_1} \tag{126}$$

*for all $s \ge 1$. This will be done in sub-subsection A.3.2.*

**Step2.** *Estimate the expression*

$$\theta_{l:k}^{(t)}(s) \overset{\text{def}}{=} \sum_{l \le i_1 < \ldots < i_s < k} \left( \frac{1}{t + i_s} + 1_{i_s - i_{s-1} = 1} \right) \cdot \left( \frac{1}{t + i_{s-1}} + 1_{i_{s-1} - i_{s-2} = 1} \right) \cdot \ldots \tag{127}$$

$$\ldots \cdot \left( \frac{1}{t + i_2} + 1_{i_2 - i_1 = 1} \right) \cdot \left( \frac{1}{t + i_1} + 1_{i_1 = 0} \right) \tag{128}$$

*for all $s \ge 1$, because as we will see the expression $\theta_{l:k}^{(t)}(s)$ can be decomposed into $\omega_{l':k'}^{(t)}(s')$ expressions for $s \ge s' \ge 1$. This will be done in sub-subsection A.3.3.*

**Step 3.** *The final step will consist in putting the results from the two previous sub-subsections together, and getting a final asymptotic estimate for $\frac{ds_{t+k}}{dh_t}$ as $k \to \infty$, by noting that*

$$\chi_{0:k}^{(t)}(s) = \frac{1}{t + k} \cdot \theta_{0:k}^{(t)}(s) + \frac{1}{t + k - 1} \cdot \theta_{0:k-1}^{(t)}(s - 1) + \ldots \tag{129}$$

$$\ldots + \frac{1}{t + k - s + 1} \cdot \theta_{0:k-s+1}^{(t)}(1) + \frac{1}{t + k - s} + 1_{k=s} \tag{130}$$

*This will be treated in sub-subsection A.3.4.*

---

### A.3.2   Estimating $\omega$

**Remark 10.** *In this sub-subsection we are going to estimate $\omega_{0:k}^{(t)}(s)$, which is a sum of products of $s$ distinct factors. The idea will be to start from the expression*

$$\left( \frac{1}{t} + \frac{1}{t + 1} + \ldots + \frac{1}{t + k - 1} \right)^s \tag{131}$$

*and substract all products containing at least two identical factors, followed by a division by $s!$.*

*This approach will be similar in spirit to the inclusion-exclusion principle, with the only difference that the desired term will not computed directly, but instead one first establishes a recursive formula using $\omega_{0:k}^{(t)}(s')$ with $s' \le s$.*

*Solving this recursive formula will enable us to express $\omega_{0:k}^{(t)}(s)$ only in terms of $(\frac{1}{t} + \frac{1}{t+1} + \ldots + \frac{1}{t+k-1})$. In fact, $\omega_{0:k}^{(t)}(s)$ will be a polynomial of degree $s$ in $(\frac{1}{t} + \frac{1}{t+1} + \ldots + \frac{1}{t+k-1})$.*

*We adopt this approach, because we have a very good estimate for*

$$\frac{1}{t} + \frac{1}{t + 1} + \ldots + \frac{1}{t + k - 1} \tag{132}$$

*Namely, we know that for all $n$, we have*

$$1 + \frac{1}{2} + \ldots + \frac{1}{n - 1} + \frac{1}{n} = \ln n + \gamma + \varepsilon_n \le \ln n + 1 \tag{133}$$

*where $\gamma > \frac{1}{2}$ is the Euler-Mascheroni constant and $\varepsilon_n$ behaves asymptotically as $\frac{1}{2n}$. In other words,*

$$\frac{1}{t} + \frac{1}{t+1} + \ldots + \frac{1}{t+k-1} = \ln\left(\frac{t+k-1}{t-1}\right) + \varepsilon_{t+k-1} - \varepsilon_{t-1} \tag{134}$$

$$= \ln\left[\frac{t+k-1}{t-1} \cdot \exp\left(\varepsilon_{t+k-1} - \varepsilon_{t-1}\right)\right] \tag{135}$$

$$= \ln \beta_{t-1,t+k-1} \tag{136}$$

*where $\beta_{l,l'} \overset{\text{def}}{=} \frac{l'}{l} \cdot \exp\left(\varepsilon_{l'} - \varepsilon_l\right)$. In order to reinforce the intuition here, let us imagine that $T = t + k$, then*

$$\ln \beta_{t-1,t+k-1} \sim \ln T \tag{137}$$

*as $T \to \infty$. Hence we should expect $\omega_{0:k}^{(t)}(s)$ to behave asymptotically as a polynomial of degree $s$ in $\ln T$.*

*Let us emphasize that we would like to express $\omega_{0:k}^{(t)}(s)$ with as much precision as possible (i.e. not omitting the monomials in $\ln T$ of degree less than $s$), since we would like to later on use this estimate in subsequent steps when summing multiple $\omega_{0:k}^{(t)}(s)$ terms over $s$.*

*In order to further ease notation, we will simply write $\omega(s)$ for $\omega_{0:k}^{(t)}(s)$, whenever there is no ambiguity.*

*Finally, for this sub-subsection only we will use the following notation*

$$S_l \overset{\text{def}}{=} \frac{1}{t^l} + \frac{1}{(t+1)^l} + \ldots + \frac{1}{(t+k-1)^l} \tag{138}$$

*for all $l \geq 1$, and keeping in mind that $S_l$ converges as $k \to \infty$, for all $l \geq 2$.*

---

**Remark 11.** *Let us now build a first intuition on how to apply an inclusion-exclusion-like principle in order to calculate $\omega(s)$ for small $s$.*

*For $\underline{s = 1}$:*

$$\omega(1) = S_1 \tag{139}$$

*For $\underline{s = 2}$:*

$$\omega(2) = \frac{1}{2!}\left(S_1^2 - S_2\right) \tag{140}$$

*Here we expand $S_1^2$, then substract the sum of products of doubles $S_2$, followed by a division of $2! = 2$ to divide out the number of permutations.*

*For $\underline{s = 3}$: first we need to substract the sum of products of triples $S_3$, and then the sum products where exactly two factors are identical $S_2 \cdot \omega(1) - S_3$. The latter appears $\binom{3}{2,1} = \frac{3!}{2!1!} = 3$ times in the expansion of $S_1^3$. Similarly, we need to divide out the number of permutations $3!$. Hence*

$$\omega(3) = \frac{1}{3!}\left[S_1^3 - S_3 - 3 \cdot (S_2 \cdot \omega(1) - S_3)\right] = \frac{S_1^3}{3!} - \frac{1}{2}S_2 \cdot \omega(1) + \frac{1}{3}S_3 \tag{141}$$

*Let us form now on denote $(3)$ for the sum of products of triples, and $(2,1)$ the sum of products where exactly two factors are the same.*

*More generally we would denote*

$$(j_1, j_2, \ldots, j_k) \tag{142}$$

*with $j_1 \geq j_2 \geq \ldots \geq j_k \geq 1$, to denote the sum of products where one factor appears exactly $j_1$ times, another factor (distinct from the previous one!) appears exactly $j_2$ times, and another factor (distinct from the previous two!) appears exactly $j_3$ times, etc. This leaves us with exactly $k$ distinct factors each having multiplicity $j_1, j_2, \ldots, j_k$ respectively. This sum appears with*

$$\binom{s}{j_1, j_2, \ldots, j_k} = \frac{s!}{j_1! \cdot j_2! \cdot \ldots \cdot j_k!} \tag{143}$$

*repetitions in the expansion of $S_1^s$, where $s = j_1 + j_2 + \ldots + j_k$.*

*For $\underline{s = 4}$: when expanding $S_1^4$, we need to take into account*

- *$(4) = S_4$ with $\binom{4}{4} = \frac{4!}{4!} = 1$ repetition.*

- *$(3,1) = S_3 \cdot \omega(1) - S_4$ with $\binom{4}{3,1} = \frac{4!}{3! \cdot 1!} = 4$ repetitions.*

- *$(2,2) = S_2^2 - S_4$ with $\binom{4}{2,2} = \frac{4!}{2! \cdot 2!} = 6$ repetitions.*

- *$(2,1,1) = S_2 \cdot \omega(2) - (3,1) = S_2 \cdot \omega(2) - S_3 \cdot \omega(1) + S_4$ with $\binom{4}{2,1,1} = \frac{4!}{2! \cdot 1! \cdot 1!} = 12$ repetitions.*

*Hence we get*

$$\omega(4) = \frac{1}{4!}[S_1^4 - S_4 - 4 \cdot (S_3 \cdot \omega(1) - S_4) - 6 \cdot (S_2^2 - S_4) \tag{144}$$

$$- 12 \cdot (S_2 \cdot \omega(2) - S_3 \cdot \omega(1) + S_4)] \tag{145}$$

$$= \frac{1}{4!}[S_1^4 - 4 \cdot S_3 \cdot \omega(1) + 4 \cdot S_4 - 6 \cdot S_2^2 + 6 \cdot S_4 - 12 \cdot S_2\omega(2) \tag{146}$$

$$+ 12 \cdot S_3 \cdot \omega(1) - 12 \cdot S_4 - S_4] \tag{147}$$

$$= \frac{1}{4!}\left[S_1^4 - 12 \cdot S_2 \cdot \omega(2) + 8 \cdot S_3 \cdot \omega(1) - 3 \cdot (S_4 + S_2^2)\right] \tag{148}$$

$$= \frac{S_1^4}{4!} - \frac{S_2}{2}\omega(2) + \frac{S_3}{3}\omega(1) - \frac{(S_4 + 2 \cdot S_2^2)}{8} \tag{149}$$

*Notice how, as we progress with higher values of $s$, we build a recursive formula in $\omega(s')$ with $s' \leq s$.*

**Intuition.** *Note that the coefficient of $\omega(2)$ for $s = 4$, is the same as the coefficient for $\omega(1)$ for $s = 3$, and is the same as the 'constant term' for $s = 2$. Similarly, the coefficient of $\omega(1)$ for $s = 4$ is the same as the 'constant term' for $s = 3$. (By convention here, we don't consider the terms $\frac{S_1^s}{s!}$ to not be part of the 'constant term'.)*

*Hence, in the recursive formula for $\omega(s)$, we would expect the coefficient of $\omega(s')$ with $s' < s$ to be equal to the 'constant term' in the formula for $\omega(s - s')$.*

**Notation.** *For all $s > l \geq 0$, let us denote $\delta_{s,l}$ to be the coefficient of the term $\omega(l)$ in the recursive formula for $\omega(s)$. By convention, we denote $\delta_{s,0}$ for the 'constant term' in the recursive formula for $\omega(s)$. Hence for all $s \geq 1$, we have*

$$\omega(s) = \frac{S_1^s}{s!} + \delta_{s,s-1} \cdot \omega(s-1) + \delta_{s,s-2} \cdot \omega(s-2) + \ldots + \delta_{s,1} \cdot \omega(1) + \delta_{s,0} \tag{150}$$

**Hypothesis.** *The hypothesis will thus rewrite as*

$$\delta_{s,l} = \delta_{s-l,0} \tag{151}$$

*for all $s > l \geq 0$, which will prove by induction on $s$ in the next lemma.*

---

**Lemma 9.** *Let $s \geq 1$. Then*

$$\omega(s) = \frac{S_1^s}{s!} + \delta_{1,0} \cdot \omega(s-1) + \delta_{2,0} \cdot \omega(s-2) + \ldots + \delta_{s-1,0} \cdot \omega(1) + \delta_{s,0} \tag{152}$$

*Proof.* Let us prove by induction on $s$ that for all $s > l \geq 0$, we have

$$\delta_{s,l} = \delta_{s-l,0} \tag{153}$$

. We already verified the cases $s = 1, 2, 3, 4$ in the previous remark. Thus let us suppose the induction hypothesis is true for $s$, and consider the mapping

$$\Upsilon : (j_1, j_2, \ldots, j_k) \mapsto (j_1, j_2, \ldots, j_k, 1) \tag{154}$$

where $j_1 \geq j_2 \geq \ldots \geq j_k \geq 1$ and $s = j_1 + j_2 + \ldots + j_k$, mapping a partition of $s$ onto a partition of $s + 1$.

If we suppose that $(j_1, j_2, \ldots, j_k)$ consists of exactly $r$ 1's, then we can write

$$(j_1, j_2, \ldots, j_k) = c_r \cdot \omega(r) + c_{r-1} \cdot \omega(r-1) + \ldots + c_1 \cdot \omega(1) + c_0 \tag{155}$$

for some coefficients $c_r, c_{r-1}, \ldots, c_1, c_0$, and with

$$\binom{s}{j_1, j_2, \ldots, j_k} = \frac{s!}{j_1! \cdot j_2 \cdot \ldots \cdot j_k!} \tag{156}$$

repetitions in the expansion of $S_1^s$.

The contribution of $(j_1, j_2, \ldots, j_k)$ to the coefficient $\delta_{s,r'}$ of $\omega(r')$ with $r' \leq r < s$, in the final recursive formula of $\omega(s)$ will be

$$\frac{c_{r'}}{j_1! \cdot j_2! \cdot \ldots \cdot j_k!} \tag{157}$$

(keeping in mind that we are dividing by $s!$ after having done all the substractions from $S_1^s$).

Meanwhile,

$$(j_1, j_2, \ldots, j_k, 1) = c_r \cdot \omega(r+1) + c_{r-1} \cdot \omega(r) + \ldots + c_1 \cdot \omega(2) + c_0 \cdot \omega(1) + \tilde{c}_0 \tag{158}$$

for some coefficient $\tilde{c}_0$, with

$$\binom{s+1}{j_1, j_2, \ldots, j_k, 1} = \frac{(s+1)!}{j_1! \cdot j_2 \cdot \ldots \cdot j_k!} \tag{159}$$

repetitions in the expansion of $S_1^{s+1}$.

The contribution of $(j_1, j_2, \ldots, j_k, 1)$ to the coefficient $\delta_{s+1,r'+1}$ of $\omega(r'+1)$ with $r' \leq r < s$, in the final recursive formula of $\omega(s+1)$ will be

$$\frac{c_{r'}}{j_1! \cdot j_2! \cdot \ldots \cdot j_k!} \tag{160}$$

(keeping in mind that we are dividing by $(s+1)!$ after having done all the substractions from $S_1^{s+1}$).

Conversely, the coefficient $\delta_{s+1,r'+1}$ only receives contributions from partitions of $(s+1)$ having at least $(r'+1)$ 1's, which correspond exactly to the contributions from the partitions of $s$ having at least $r'$ 1's. Hence

$$\delta_{s+1,r'+1} = \delta_{s,r'} \tag{161}$$

Then by the induction hypothesis, we have $\delta_{s,r'} = \delta_{s-r',0}$. In other words

$$\delta_{s+1,r'+1} = \delta_{s-r',0} \tag{162}$$

which completes the proof by induction.

$\square$

**Remark 12.** *Note that all the coefficients $\delta_{s,l}$ consist of linear combination of products with factors equal to $S_j$ with $j \geq 2$, which are known to converge as $T \to \infty$. Thus those can be considered constants when doing an asymptotic analysis in the subsequent sub-subsections. Also note that $\delta_{s,s-1} = \delta_{1,0} = 0$.*

**Proposition 3.** *For all $s \geq 1$, we have*

$$\omega(s) = \sum_{r=0}^{s} \psi_{s-r} \frac{S_1^r}{r!} \tag{163}$$

*where for $l \geq 2$*

$$\psi_l \stackrel{\text{def}}{=} \sum_{k=1}^{l-1} \sum_{(j_1,j_2,\ldots,j_k) \in \Psi_{l,k}} \delta_{j_1,0} \cdot \ldots \cdot \delta_{j_k,0} \tag{164}$$

*with*

$$\Psi_{l,k} \stackrel{\text{def}}{=} \{(j_1,j_2,\ldots,j_k) \text{ with } j_1 \geq \ldots \geq j_k > 1 \text{ and } j_1 + \ldots + j_k = l\} \tag{165}$$

*and where we define $\psi_0 = 1$ and $\psi_1 = 0$.*

*Proof.* For $l \geq 2$, we have

$$\psi_l = \sum_{k=1}^{l-1} \sum_{(j_1,j_2,\ldots,j_k) \in \Psi_{l,k}} \delta_{j_1,0} \cdot \ldots \cdot \delta_{j_k,0} \tag{166}$$

$$= \delta_{l,0} + \sum_{k=1}^{l-1} \left( \sum_{j=2}^{l-2} \sum_{(j_2,\ldots,j_k) \in \Psi_{l-j,k-1}} \delta_{j,0} \cdot \delta_{j_2,0} \cdot \ldots \cdot \delta_{j_k,0} \right) \tag{167}$$

$$= \delta_{l,0} + \sum_{j=2}^{l-2} \left( \sum_{k=1}^{l-j} \sum_{(j_2,\ldots,j_k) \in \Psi_{l-j,k-1}} \delta_{j,0} \cdot \delta_{j_2,0} \cdot \ldots \cdot \delta_{j_k,0} \right) \tag{168}$$

$$= \delta_{l,0} + \sum_{j=2}^{l-2} \delta_{j,0} \cdot \left( \sum_{k=1}^{l-j} \sum_{(j_2,\ldots,j_k) \in \Psi_{l-j,k-1}} \delta_{j_2,0} \cdot \ldots \cdot \delta_{j_k,0} \right) \tag{169}$$

$$= \delta_{l,0} + \sum_{j=2}^{l-2} \delta_{j,0} \cdot \psi_{l-j} \tag{170}$$

$$= \sum_{j=1}^{l} \delta_{j,0} \cdot \psi_{l-j} \tag{171}$$

$$\tag{172}$$

In other words, we have shown that for all $l \geq 2$,

$$\psi_l = \sum_{j=0}^{l-1} \delta_{l-j} \psi_j \tag{173}$$

Let us now prove the proposition by induction on $s$.

The case $\underline{s=1}$ is trivial by the definition of $\psi_0$ and $\psi_1$.

Let us now assume the formula is true for $s$, and let us prove it for $s+1$. By the previous lemma 3, we know that

$$\omega(s+1) = \frac{S_1^{s+1}}{(s+1)!} + \sum_{l=1}^{s} \delta_{s+1-l,0} \cdot \omega(l) + \delta_{s+1,0} \tag{174}$$

$$= \frac{S_1^{s+1}}{(s+1)!} + \sum_{l=1}^{s} \delta_{s+1-l,0} \cdot \left( \sum_{r=0}^{l} \psi_{l-r} \cdot \frac{S_1^r}{r!} \right) + \delta_{s+1,0} \tag{175}$$

$$= \frac{S_1^{s+1}}{(s+1)!} + \sum_{l=1}^{s} \sum_{r=0}^{l} \delta_{s+1-l,0} \cdot \psi_{l-r} \cdot \frac{S_1^r}{r!} + \delta_{s+1,0} \tag{176}$$

$$= \frac{S_1^{s+1}}{(s+1)!} + \sum_{l=0}^{s} \sum_{r=0}^{l} \delta_{s+1-l,0} \cdot \psi_{l-r} \cdot \frac{S_1^r}{r!} \tag{177}$$

$$= \frac{S_1^{s+1}}{(s+1)!} + \sum_{r=0}^{s} \sum_{l=r}^{s} \delta_{s+1-l,0} \cdot \psi_{l-r} \cdot \frac{S_1^r}{r!} \tag{178}$$

$$= \frac{S_1^{s+1}}{(s+1)!} + \sum_{r=0}^{s} \sum_{l'=0}^{s-r} \delta_{s+1-r-l',0} \cdot \psi_{l'} \cdot \frac{S_1^r}{r!} \tag{179}$$

$$= \frac{S_1^{s+1}}{(s+1)!} + \sum_{r=0}^{s} \psi_{s+1-r} \cdot \frac{S_1^r}{r!} \tag{180}$$

$$= \sum_{r=0}^{s+1} \psi_{s+1-r} \cdot \frac{S_1^r}{r!} \tag{181}$$

completing the proof by induction. $\qquad\square$

---

**Remark 13.** *Hence we have shown that for all $s \geq 1$*

$$\omega(s) = \sum_{r=0}^{s} \psi_{s-r} \frac{S_1^r}{r!} = \frac{S_1^s}{s!} + \sum_{r=0}^{s-2} \psi_{s-r} \frac{S_1^r}{r!} \tag{182}$$

*or in other words*

$$\omega(s) = \frac{(\ln \beta_{t-1,t+k-1})^s}{s!} + \sum_{r=0}^{s-2} \psi_{s-r} \frac{(\ln \beta_{t-1,t+k-1})^r}{r!} \sim \frac{(\ln T)^s}{s!} + \sum_{r=0}^{s-2} \psi_{s-r} \frac{(\ln T)^r}{r!} \tag{183}$$

*as $t+k = T \to \infty$, which is roughly the polynomial in $\ln T$ of degree $s$ we were anticipating.*

---

### A.3.3 Estimating $\theta$

**Remark 14.** *Let us now recall the definition for all $s \geq 1$,*

$$\theta_{l:k}^{(t)}(s) \stackrel{\text{def}}{=} \sum_{l \leq i_1 < \ldots < i_s < k} \left( \frac{1}{t+i_s} + 1_{i_s - i_{s-1} = 1} \right) \cdot \left( \frac{1}{t+i_{s-1}} + 1_{i_{s-1} - i_{s-2} = 1} \right) \cdot \ldots \tag{184}$$

$$\ldots \cdot \left( \frac{1}{t+i_2} + 1_{i_2 - i_1 = 1} \right) \cdot \left( \frac{1}{t+i_1} + 1_{i_1 = 0} \right) \tag{185}$$

*which we would like to estimate using $\omega_{l:k}^{(t)}(s)$.*

*In order to build a first intuition, let us look at how it plays out for small values for $s$.*

**Notation.** *In this subsection we omit the superscript $(t)$ notation because there is no ambiguity. We will also occasionally do the abuse of notation and assume $\omega_{l:k}(0) = 1$ for all $l < k$.*

*For $\underline{s = 1}$, we get*

$$\theta_{0:k}(1) = 1 + \omega_{0:k}(1) \tag{186}$$

*For $\underline{s = 2}$, we get*

$$\theta_{0:k}(2) = 1 + \omega_{1:k}(1) + \omega_{0:k-1}(1) + \omega_{0:k}(2) \tag{187}$$

*In what follows, we will use the following recursive formula quite frequently*

$$\theta_{0:k}(s+1) = \theta_{0:k-1}(s) + \sum_{j=s}^{k-1} \frac{1}{t+j} \theta_{0:j}(s) \tag{188}$$

*Hence for $\underline{s = 3}$, we get*

$$\theta_{0:k}(3) = 1 + \omega_{1:k-1}(1) + \omega_{0:k-2}(1) + \omega_{2:k}(1) + \omega_{0:k-1}(2) \tag{189}$$

$$+ \omega_{1:k}(2) + \sum_{j=2}^{k-1} \frac{\omega_{0:j-1}(1)}{t+j} + \omega_{0:k}(3) \tag{190}$$

*Now let us further observe that for all $s \geq 1$ and $0 \leq r \leq l$, we have*

$$\omega_{l+r:k+r}(s) \leq \omega_{l:k}(s) \leq \omega_{l-r:k-r}(s) \tag{191}$$

*This implies that*

$$1 + 2 \cdot \omega_{1:k}(1) + \omega_{0:k}(2) \leq \theta_{0:k}(2) \leq 1 + 2 \cdot \omega_{0:k-1}(1) + \omega_{0:k}(2) \tag{192}$$

*and, similarly,*

$$1 + 3 \cdot \omega_{2:k}(1) + 3 \cdot \omega_{1:k}(2) + \omega_{0:k}(3) \leq \theta_{0:k}(3) \leq 1 + 3 \cdot \omega_{0:k-2}(1) + 3 \cdot \omega_{0:k-1}(2) + \omega_{0:k}(3) \tag{193}$$

**Hypothesis.** *We can thus see the binomial coefficients arising, and we would expect that in general, we have*

$$\sum_{r=0}^{s} \binom{s}{r} \cdot \omega_{0:k-s+r}(r) \geq \theta_{0:k}(s) \geq \sum_{r=0}^{s} \binom{s}{r} \cdot \omega_{s-r:k}(r) \tag{194}$$

---

**Lemma 10.** *For all $k \geq s \geq 1$, we have*

$$\sum_{r=0}^{s} \binom{s}{r} \cdot \omega_{0:k-s+r}^{(t)}(r) \geq \theta_{0:k}^{(t)}(s) \geq \sum_{r=0}^{s} \binom{s}{r} \cdot \omega_{s-r:k}^{(t)}(r) \tag{195}$$

*Proof.* Let us prove this lemma by induction on $s$. The cases $s = 1, 2, 3$ have already been treated in the previous remark.

Let us now assume that the claim holds for $s$, and prove it for $s + 1$ using the recursive formula

$$\theta_{0:k}(s+1) = \theta_{0:k-1}(s) + \sum_{j=s}^{k-1} \frac{1}{t+j} \theta_{0:j}(s) \tag{196}$$

For the lower bound, using the induction hypothesis, we get

$$\theta_{0:k}(s+1) \geq \sum_{r=0}^{s} \binom{s}{r} \cdot \omega_{s-r:k-1}(r) + \sum_{j=s}^{k-1} \frac{1}{t+j} \sum_{r=0}^{s} \binom{s}{r} \cdot \omega_{s-r:j}(r) \tag{197}$$

$$= \sum_{r=0}^{s} \binom{s}{r} \cdot \omega_{s-r:k-1}(r) + \sum_{r=0}^{s} \binom{s}{r} \cdot \sum_{j=s}^{k-1} \frac{1}{t+j} \cdot \omega_{s-r:j}(r) \tag{198}$$

$$= \sum_{r=0}^{s} \binom{s}{r} \cdot \omega_{s-r:k-1}(r) + \sum_{r=0}^{s} \binom{s}{r} \cdot \omega_{s-r:k}(r+1) \tag{199}$$

$$= \sum_{r=0}^{s} \binom{s}{r} \cdot \omega_{s-r:k-1}(r) + \sum_{r=1}^{s+1} \binom{s}{r-1} \cdot \omega_{s-r+1:k}(r) \tag{200}$$

$$= 1 + \omega_{0:k}(s+1) + \sum_{r=1}^{s} \left[ \binom{s}{r} + \binom{s}{r-1} \right] \cdot \omega_{s-r+1:k}(r) \tag{201}$$

$$= 1 + \omega_{0:k}(s+1) + \sum_{r=1}^{s} \binom{s+1}{r} \cdot \omega_{s-r+1:k}(r) \tag{202}$$

$$= \sum_{r=0}^{s+1} \binom{s+1}{r} \cdot \omega_{s-r+1:k}(r) \tag{203}$$

$$\tag{204}$$

For the upper bound, using the induction hypothesis, we get

$$\theta_{0:k}(s+1) \leq \sum_{r=0}^{s} \binom{s}{r} \cdot \omega_{0:k-1-(s-r)}(r) + \sum_{j=s}^{k-1} \frac{1}{t+j} \sum_{r=0}^{s} \binom{s}{r} \cdot \omega_{0:j-(s-r)}(r) \tag{205}$$

$$= \sum_{r=0}^{s} \binom{s}{r} \cdot \omega_{0:k-1-(s-r)}(r) + \sum_{r=0}^{s} \binom{s}{r} \cdot \sum_{j=s}^{k-1} \frac{1}{t+j} \cdot \omega_{0:j-(s-r)}(r) \tag{206}$$

$$\leq \sum_{r=0}^{s} \binom{s}{r} \cdot \omega_{0:k-1-(s-r)}(r) + \sum_{r=0}^{s} \binom{s}{r} \cdot \sum_{j=s}^{k-1} \frac{1}{t+j-(s-r)} \cdot \omega_{0:j-(s-r)}(r) \tag{207}$$

$$= \sum_{r=0}^{s} \binom{s}{r} \cdot \omega_{0:k-1-(s-r)}(r) + \sum_{r=0}^{s} \binom{s}{r} \cdot \sum_{j'=r}^{k-1-(s-r)} \frac{1}{t+j'} \cdot \omega_{0:j'}(r) \tag{208}$$

$$= \sum_{r=0}^{s} \binom{s}{r} \cdot \omega_{0:k-1-(s-r)}(r) + \sum_{r=0}^{s} \binom{s}{r} \cdot \omega_{0:k-(s-r)}(r+1) \tag{209}$$

$$= \sum_{r=0}^{s} \binom{s}{r} \cdot \omega_{0:k-1-(s-r)}(r) + \sum_{r=1}^{s+1} \binom{s}{r-1} \cdot \omega_{0:k-(s+1-r)}(r) \tag{210}$$

$$= 1 + \omega_{0:k}(s+1) + \sum_{r=1}^{s} \left[ \binom{s}{r} + \binom{s}{r-1} \right] \cdot \omega_{0:k-(s+1-r)}(r) \tag{211}$$

$$= 1 + \omega_{0:k}(s+1) + \sum_{r=1}^{s} \binom{s+1}{r} \cdot \omega_{0:k-(s+1-r)}(r) \tag{212}$$

$$= \sum_{r=0}^{s+1} \binom{s+1}{r} \cdot \omega_{0:k-(s+1-r)}(r) \tag{213}$$

$$\tag{214}$$

completing the proof by induction. □

**Remark 15.** *Let us recall that*

$$\omega_{l:k}(r) = \sum_{q=0}^{r} \psi_{r-q} \frac{(\ln \beta_{t+l-1,t+k-1})^q}{q!} \tag{215}$$

*Thus the difference between the upper-bound and the lower-bound becomes*

$$\sum_{r=0}^{s} \binom{s}{r} \left[ \omega_{o:k-(s-r)}(r) - \omega_{s-r:k}(r) \right] = \sum_{r=0}^{s} \binom{s}{r} \cdot \left[ \sum_{q=0}^{r} \psi_{r-q} \frac{(\ln \beta_{t-1,t+k-(s-r)-1})^q - (\ln \beta_{t+s-r-1,t+k-1})^q}{q!} \right] \tag{216}$$

*which converges to zero as $T = t + k \to \infty$.*

---

### A.3.4 Putting it all together

**Remark 16.** *Now it is time to turn to $\chi_{0:k}^{(t)}(s)$ and finally put it all together, so that we can finally estimate*

$$\frac{ds_{t+k}}{dh_t} = \sum_{s=0}^{k} V^s \cdot \chi_{0:k}^{(t)}(s) \tag{217}$$

*and get the asymptotic estimate when $T = t + k \to \infty$.*

*Let us recall that*

$$\chi_{0:k}^{(t)}(s) = \frac{1}{t+k} \cdot \theta_{0:k}^{(t)}(s) + \frac{1}{t+k-1} \cdot \theta_{0:k-1}^{(t)}(s-1) + \ldots \tag{218}$$

$$\ldots + \frac{1}{t+k-s+1} \cdot \theta_{0:k-s+1}^{(t)}(1) + \frac{1}{t+k-s} + 1_{k=s} \tag{219}$$

*Using the abuse of notation $\theta_{l:k}(0) = 1$ for $l < k$, we can rewrite it as follows*

$$\chi_{0:k}^{(t)}(s) = 1_{k=s} + \sum_{i=0}^{s} \frac{1}{t+k-i} \cdot \theta_{0:k-i}(s-i) \tag{220}$$

*The idea is to use the inequality from lemma 10, and get a similar result for $\chi_{0:k}^{(t)}(s)$, then show that the lower and upper bound are no more than $\Theta(1/T)$ apart, thus enabling us to eventually get an asymptotic estimate for $\frac{ds_{t+k}}{dh_t}$.*

*We are also omitting the superscript $(t)$ notation here because of lack of ambiguity.*

---

**Lemma 11.** *For all $s \geq 0$ and $k \geq 1$, we have*

$$1_{k=s} + \frac{1}{t+k} \cdot \sum_{r=0}^{s} \binom{s+1}{r+1} \cdot \omega_{s-r:k}(r) \leq \chi_{0:k}(s) \leq 1_{k=s} + \frac{1}{t+k-s} \cdot \sum_{r=0}^{s} \binom{s+1}{r+1} \cdot \omega_{0:k-(s-r)}(r) \tag{221}$$

*Proof.* Using the upper-bound of lemma 10, we get

$$\chi_{0:k}(s) = 1_{k=s} + \sum_{i=0}^{s} \frac{1}{t+k-i} \cdot \theta_{0:k-i}(s-i) \tag{222}$$

$$\leq 1_{k=s} + \sum_{i=0}^{s} \frac{1}{t+k-i} \cdot \sum_{r=0}^{s-i} \binom{s-i}{r} \cdot \omega_{0:k-s+r}(r) \tag{223}$$

$$\leq 1_{k=s} + \frac{1}{t+k-s} \cdot \sum_{i=0}^{s} \sum_{r=0}^{s-i} \binom{s-i}{r} \cdot \omega_{0:k-s+r}(r) \tag{224}$$

$$= 1_{k=s} + \frac{1}{t+k-s} \cdot \sum_{r=0}^{s} \left[ \sum_{i=0}^{s-r} \binom{s-i}{r} \right] \cdot \omega_{0:k-s+r}(r) \tag{225}$$

$$= 1_{k=s} + \frac{1}{t+k-s} \cdot \sum_{r=0}^{s} \binom{s+1}{r+1} \cdot \omega_{0:k-s+r}(r) \tag{226}$$

Similarly, using the lower-bound of lemma 10, we get

$$\chi_{0:k}(s) = 1_{k=s} + \sum_{i=0}^{s} \frac{1}{t+k-i} \cdot \theta_{0:k-i}(s-i) \tag{227}$$

$$\geq 1_{k=s} + \sum_{i=0}^{s} \frac{1}{t+k-i} \cdot \sum_{r=0}^{s-i} \binom{s-i}{r} \cdot \omega_{s-r:k}(r) \tag{228}$$

$$\geq 1_{k=s} + \frac{1}{t+k} \cdot \sum_{i=0}^{s} \sum_{r=0}^{s-i} \binom{s-i}{r} \cdot \omega_{s-r:k}(r) \tag{229}$$

$$= 1_{k=s} + \frac{1}{t+k} \cdot \sum_{r=0}^{s} \left[ \sum_{i=0}^{s-r} \binom{s-i}{r} \right] \cdot \omega_{s-r:k}(r) \tag{230}$$

$$= 1_{k=s} + \frac{1}{t+k} \cdot \sum_{r=0}^{s} \binom{s+1}{r+1} \cdot \omega_{s-r:k}(r) \tag{231}$$

$$\square$$

---

**Lemma 12.** *For all $s \geq 0$, we have*

$$\chi_{0:k}(s) = 1_{k=s} + \frac{1}{t+k} \left[ \sum_{r=0}^{s} \binom{s+1}{r+1} \cdot \omega_{s-r:k}(r) \right] + \Theta\left( \frac{1}{t+k} \right) \tag{232}$$

*for all large enough $k > 1$, and where the implicit constants from the $\Theta(.)$ notation are dependent on $s$.*

*Proof.* Building on the previous lemma 11, and substracting the lower bound from the upper bound, we get

$$\sum_{r=0}^{s} \binom{s+1}{r+1} \cdot \left[ \frac{\omega_{0:k-s+r}(r)}{t+k-s} - \frac{\omega_{s-r:k}(r)}{t+k} \right] = \sum_{r=0}^{s} \sum_{q=0}^{r} \binom{s+1}{r+1} \frac{\psi_{r-q}}{q!} \cdot \left[ \frac{(\ln \beta_{t-1,t+k-s+r-1})^q}{t+k-s} - \frac{(\ln \beta_{t+s-r-1,t+k-1})^q}{t+k} \right] \tag{233}$$

When assuming that for large $k$, we have

$$(\ln \beta_{t-1,t+k-s+r-1})^q \approx (\ln \beta_{t+s-r-1,t+k-1})^q \tag{234}$$

then

$$\frac{(\ln \beta_{t-1,t+k-s+r-1})^q}{t+k-s} - \frac{(\ln \beta_{t+s-r-1,t+k-1})^q}{t+k} \approx \frac{1}{t+k} \cdot \left[ \frac{s}{t+k-s} \cdot (\ln \beta_{t-1,t+k-s+r-1})^q \right] \tag{235}$$

$$\leq \frac{1}{t+k} \cdot \left[ \frac{s}{t+k-s} \cdot (\ln \beta_{t-1,t+k-s+r-1})^s \right] \tag{236}$$

$$\leq \frac{\tau_s}{t+k} \tag{237}$$

for some $\tau_s > 0$ depending on $s$, for all sufficiently large $k$.

In other words, we have

$$\sum_{r=0}^{s} \binom{s+1}{r+1} \cdot \left[ \frac{\omega_{0:k-s+r}(r)}{t+k-s} - \frac{\omega_{s-r:k}(r)}{t+k} \right] \leq \frac{\tilde{\tau}_s}{t+k} \tag{238}$$

for for some $\tilde{\tau}_s > 0$ depending on $s$, for all sufficiently large $k$.

Meanwhile, for all large enough $k$, we have

$$\sum_{r=0}^{s} \binom{s+1}{r+1} \cdot \left[ \frac{\omega_{0:k-s+r}(r)}{t+k-s} - \frac{\omega_{s-r:k}(r)}{t+k} \right] \approx \frac{s}{(t+k)(t+k-s)} \cdot \sum_{r=0}^{s} \sum_{q=0}^{r} \binom{s+1}{r+1} \frac{\psi_{r-q}}{q!} \cdot (\ln \beta_{t-1,t+k-s+r-1})^q \tag{239}$$

$$\geq \frac{\tau'_s}{(t+k)^2} \cdot \sum_{r=0}^{s} \sum_{q=0}^{r} \binom{s+1}{r+1} \frac{\psi_{r-q}}{q!} \cdot (\ln \beta_{t-1,t+k-s+r-1})^q \tag{240}$$

$$\geq \frac{\tau'_s}{(t+k)^2} \cdot \sum_{r=0}^{s} \sum_{q=0}^{r} \frac{\psi_{r-q}}{q!} \cdot (\ln \beta_{t-1,t+k-s+r-1})^q \tag{241}$$

$$= \frac{\tau'_s}{(t+k)^2} \cdot \sum_{q=0}^{s} \sum_{r'=0}^{s-q} \frac{\psi_{r'}}{q!} \cdot (\ln \beta_{t-1,t+k-s+r'+q-1})^q \tag{242}$$

$$\approx \frac{\tau'_s}{(t+k)^2} \cdot \sum_{q=0}^{s} \left( \sum_{r'=0}^{s-q} \psi_{r'} \right) \cdot \frac{(\ln (t+k))^q}{q!} \tag{243}$$

$$\geq \frac{\tau''_s}{(t+k)^2} \cdot \sum_{q=0}^{s} \frac{(\ln (t+k))^q}{q!} \tag{244}$$

$$\approx \frac{\tau''_s \cdot \exp[\ln (t+k)]}{(t+k)^2} \tag{245}$$

$$= \frac{\tau''_s}{t+k} \tag{246}$$

for some $\tau'_s, \tau''_s > 0$ depending on $s$. $\qquad \square$

---

**Proposition 4.** *If $V$ is a normal matrix with eigenvalues $\lambda_1, \lambda_2, \ldots, \lambda_n$ of modulus smaller than 1, then*

$$\frac{ds_T}{dh_t} = P\Lambda_T P^* \tag{247}$$

*where $P^*$ is the conjugate transpose of the unitary matrix $P$ (independent of $T$) and where $\Lambda_T$ is a diagonal matrix satisfying*

$$(\Lambda_T)_{ii} \sim T^{-1} \cdot c + T^{\lambda_i - 1} \cdot c' \tag{248}$$

*for some positive real constants $c, c'$, as $T \to \infty$.*

*Proof.* Let $V = P\Lambda P^*$ be the Schur decomposition of $V$, with $\Lambda = \text{diag}(\lambda_1, \lambda_2, \ldots, \lambda_n)$. Note that since we supposed that $V$ is normal, we thus have that the Schur matrix $\Lambda$ is indeed diagonal and is composed of the eigenvalues on the diagonal.

Based on lemma 12, one can show that there exists a function $g : \mathbb{N} \to \mathbb{R}_0^+$ such that

$$\chi_{0:k}(s) = 1_{k=s} + \frac{1}{t+k} \left[ \sum_{r=0}^{s} \binom{s+1}{r+1} \cdot \omega_{s-r:k}(r) + g(s) \right] \tag{249}$$

Thus

$$\frac{ds_{t+k}}{dh_t} = \sum_{s=0}^{k} V^s \cdot \chi_{0:k}(s) \tag{250}$$

$$= V^k + \frac{1}{t+k} \left[ \sum_{s=0}^{k} g(s) \cdot V^s + \sum_{s=0}^{k} \sum_{r=0}^{s} \binom{s+1}{r+1} \cdot \omega_{s-r:k}(r) \cdot V^s \right] \tag{251}$$

$$= V^k + \frac{1}{t+k} \left[ \sum_{s=0}^{k} g(s) \cdot V^s + \sum_{s=0}^{k} \sum_{r=0}^{s} \sum_{q=0}^{r} \binom{s+1}{r+1} \cdot \psi_{r-q} \frac{(\ln \beta_{t+s-r-1,t+k-1})^q}{q!} \cdot V^s \right] \tag{252}$$

$$\tag{253}$$

Since the eigenvalues of $V$ are of modulus smaller than 1, we can assume that there exists a constant $d > 0$ (dependent on the choice of eigenvalues of $V$) such that for all $k > d$ we have $V^k \approx 0$.

Furthermore since $V^m = (P\Lambda P^*)^m = P\Lambda^m P^*$ for all $m \in \mathbb{N}_0$, while keeping in mind that we pick $T = t + k$, we can write

$$\Lambda_T = \frac{1}{T} \left[ \sum_{s=0}^{d} g(s) \cdot \Lambda^s + \sum_{s=0}^{d} \sum_{r=0}^{s} \sum_{q=0}^{r} \binom{s+1}{r+1} \cdot \psi_{r-q} \frac{(\ln \beta_{t+s-r-1,T-1})^q}{q!} \cdot \Lambda^s \right] \tag{254}$$

$$= \frac{1}{T} \left[ \sum_{s=0}^{d} g(s) \cdot \Lambda^s + \sum_{s=0}^{d} \sum_{q=0}^{s} \sum_{r=q}^{s} \binom{s+1}{r+1} \cdot \psi_{r-q} \frac{(\ln \beta_{t+s-r-1,T-1})^q}{q!} \cdot \Lambda^s \right] \tag{255}$$

$$= \frac{1}{T} \left[ \sum_{s=0}^{d} g(s) \cdot \Lambda^s + \sum_{q=0}^{d} \sum_{s=q}^{d} \sum_{r=q}^{s} \binom{s+1}{r+1} \cdot \psi_{r-q} \frac{(\ln \beta_{t+s-r-1,T-1})^q}{q!} \cdot \Lambda^s \right] \tag{256}$$

$$= \frac{1}{T} \left[ \sum_{s=0}^{d} g(s) \cdot \Lambda^s + \sum_{q=0}^{d} \sum_{s=q}^{d} \sum_{r=q}^{s} \binom{s+1}{r+1} \cdot \psi_{r-q} \frac{(\Lambda \cdot \ln \beta_{t+s-r-1,T-1})^q}{q!} \cdot \Lambda^{s-q} \right] \tag{257}$$

$$= \frac{1}{T} \left[ \sum_{s=0}^{d} g(s) \cdot \Lambda^s + \sum_{q=0}^{d} \sum_{s'=0}^{d-q} \sum_{r'=0}^{s'} \binom{s'+q+1}{r'+q+1} \cdot \psi_{r'} \frac{(\Lambda \cdot \ln \beta_{t+s'-r'-1,T-1})^q}{q!} \cdot \Lambda^{s'} \right] \tag{258}$$

$$\sim \frac{1}{T} \left[ \sum_{s=0}^{d} g(s) \cdot \Lambda^s + \sum_{q=0}^{d} \sum_{s'=0}^{d-q} \sum_{r'=0}^{s'} \binom{s'+q+1}{r'+q+1} \cdot \psi_{r'} \frac{(\Lambda \cdot \ln T)^q}{q!} \cdot \Lambda^{s'} \right] \tag{259}$$

$$= \frac{1}{T} \left[ \sum_{s=0}^{d} g(s) \cdot \Lambda^s \right] + \frac{1}{T} \left[ \sum_{q=0}^{d} \frac{(\Lambda \cdot \ln T)^q}{q!} \cdot \left( \sum_{s'=0}^{d-q} \sum_{r'=0}^{s'} \binom{s'+q+1}{r'+q+1} \cdot \psi_{r'} \cdot \Lambda^{s'} \right) \right] \tag{260}$$

$$\approx \frac{1}{T} \left[ \sum_{s=0}^{d} g(s) \cdot \Lambda^s \right] + \frac{1}{T} \exp\left( \Lambda \cdot \ln T \right) \cdot (c_0 + c_1 \cdot \Lambda + \ldots + c_d \cdot \Lambda^d) \tag{261}$$

$$\sim \frac{c}{T} + \frac{c'}{T} \exp\left( \Lambda \cdot \ln T \right) \tag{262}$$

for some positive constants $c', c, c_0, c_1, \ldots, c_d$.

Hence

$$(\Lambda_T)_{ii} \sim c \cdot T^{-1} + c' \cdot T^{\lambda_i - 1} \tag{263}$$

$\square$

---

**Theorem 1.** *If $V$ is a normal matrix with eigenvalues of modulus smaller than $1$, then*

$$\|\frac{ds_T}{dh_t}\| = \Omega(1/T) \tag{264}$$

*as $T \to \infty$. (here $\|.\|$ is the Frobenius norm.)*

*Proof.* Let us start off with the observation that

$$T^{-1} \cdot c + T^{\lambda_i - 1} \cdot c' = \Omega\left(T^{-\min(1, 1 - \mathfrak{Re}(\lambda_i))}\right) \tag{265}$$

as $T \to \infty$. And thus, by using proposition 4, we get

$$\|\frac{ds_T}{dh_t}\| = \Omega(T^{-\eta}) \tag{266}$$

where

$$\eta = \min_{i=1,\dots,n} \{\min(1, 1 - \mathfrak{Re}(\lambda_i))\} \leq 1 \tag{267}$$

$\square$

---

**Remark 17.** *Note that $V$ being normal is not a necessary condition for the generality of the theorem to hold. We simply chose $V$ to be normal in order to make the calculations less cumbersome.*

*In case $V$ is non-normal, its Schur matrix $\Lambda$ becomes triangular instead of diagonal. In fact, if $t_{i,j}$ are the off-diagonal elements of Schur matrix of $V$ (with $i < j$), then*

$$\|V\| = \sqrt{Tr(V^*V)} = \sqrt{Tr(\Lambda^*\Lambda)} = \sqrt{\sum_{i=1}^n |\lambda_i|^2 + \sum_{i<j} |t_{i,j}|^2} \geq \sqrt{\sum_{i=1}^n |\lambda_i|^2} \tag{268}$$

*Thus every lower bound on $\sqrt{\sum_{i=1}^n |\lambda_i|^2}$ induces a lower bound on $\|V\|$, and in particular an asymptotic lower bound on the modulus of one of the eigenvalues of $\frac{ds_T}{dh_t}$ induces an asymptotic lower bound on $\|\frac{ds_T}{dh_t}\|$.*

---

### A.4 Sparse relevance case with bounded dependency depth

**Remark 18.** *Similarly to remark 7, we are going to assume for this subsection:*

- *no non-linearity in the hidden-to-hidden connection: $J_t = V$ for all $t$.*

- *all assumptions from Remark 3.*

- *$\kappa$-sparse attention: for each $t \geq 1$, there are at most $\kappa \leq t$ values for $i$ such that $\alpha_{i,t} \neq 0$. (Let us define $\kappa_t \overset{\text{def}}{=} |\{i \text{ such that } \alpha_{i,t} \neq 0\}|$)*

- *uniform attention across attended states: for all $t \geq 1$, and all $i \leq t$ such that $\alpha_{i,t} \neq 0$, we have $\alpha_{i,t} = 1/\kappa_t \geq 1/\kappa$.*

---

**Remark 19.** *Similarly to remark 8, let us recall that*

$$X_{i,t} = \left( h_i - \sum_{j=1}^{t} \alpha_{j,t} h_j \right) \cdot \frac{\partial e_{i,t}}{\partial h_i} \tag{269}$$

*and that*

$$\sum_{i=1}^{t} \alpha_{i,t} Y_{i,t} = \sum_{i=1}^{t} \alpha_{i,t} \cdot h_i \cdot \left( \frac{\partial e_{i,t}}{\partial s_{t-1}} - \sum_{j=1}^{t} \alpha_{j,t} \cdot \frac{\partial e_{j,t}}{\partial s_{t-1}} \right) \tag{270}$$

$$= \sum_{i=1}^{t} \alpha_{i,t} \cdot h_i \cdot \frac{\partial e_{i,t}}{\partial s_{t-1}} - \sum_{i=1}^{t} \alpha_{i,t} \left( \sum_{j=1}^{t} \alpha_{j,t} \cdot h_j \right) \cdot \frac{\partial e_{i,t}}{\partial s_{t-1}} \tag{271}$$

$$= \sum_{i=1}^{t} \alpha_{i,t} \cdot \left( h_i - \sum_{j=1}^{t} \alpha_{j,t} h_j \right) \cdot \frac{\partial e_{i,t}}{\partial s_{t-1}} \tag{272}$$

*Thus we can see that both expressions have the common factor $\left( h_i - \sum_{j=1}^{t} \alpha_{j,t} h_j \right)$.*

*By defining*

$$A_t \overset{\text{def}}{=} \{i \text{ such that } \alpha_{i,t} \neq 0\} \tag{273}$$

*we see that*

$$h_i - \sum_{j=1}^{t} \alpha_{j,t} h_j = h_i - \frac{1}{\kappa_t} \sum_{j \in A_t} h_j \tag{274}$$

*and we are going to assume for the sake of simplicity that*

$$h_i \approx \frac{1}{\kappa_t} \sum_{j \in A_t} h_j \tag{275}$$

*and thus $X_{i,t} \approx 0$ and $\sum_{i=1}^{t} \alpha_{i,t} Y_{i,t} \approx 0$.*

*Recalling the expression from corollary 1 and that $f(h_t, c_t) = h_t + c_t$ by remark 3, and that $J_t = V$ for all t, this will give for all $k' \geq 0$*

$$E_{k'}^{(t)} = \left( \frac{1_{t \in A_{t+k'}}}{\kappa_{t+k'}} + 1_{k'=0} \right) \cdot I \tag{276}$$

*and for all $k \geq j$, we get*

$$F_{k+1,j}^{(t)} = \left( \frac{1_{t+j+1 \in A_{t+k+1}}}{\kappa_{t+k+1}} + 1_{k=j} \right) \cdot V \tag{277}$$

*Hence by recalling proposition 1, the main expression of interest becomes*

$$\frac{ds_{t+k}}{dh_t} = \sum_{s=0}^{k} \bar{\xi}_{0:k}^{(t)}(s) = \sum_{s=0}^{k} V^s \cdot \chi_{0:k}^{(t)}(s) \tag{278}$$

*where*

$$\chi_{0:k}^{(t)}(s) \overset{\text{def}}{=} \sum_{0 \leq i_1 < \ldots < i_s < k} \left( \frac{1_{t+i_s+1 \in A_{t+k}}}{\kappa_{t+k}} + 1_{k-i_s=1} \right) \cdot \left( \frac{1_{t+i_{s-1}+1 \in A_{t+i_s}}}{\kappa_{t+i_s}} + 1_{i_s - i_{s-1}=1} \right) \cdot \tag{279}$$

$$\ldots \cdot \left( \frac{1_{t+i_1+1 \in A_{t+i_2}}}{\kappa_{t+i_2}} + 1_{i_2 - i_1=1} \right) \cdot \left( \frac{1_{t \in A_{t+i_1}}}{\kappa_{t+i_1}} + 1_{i_1=0} \right) \tag{280}$$

**Remark 20.** *Let us now have a look at how we could potentially simplify the analysis of $\chi_{0:k}^{(t)}(s)$. If we further assume $V$ to be normal we can write*

$$V = P\Lambda P^* \tag{281}$$

*where $\Lambda = diag(\lambda_1, \lambda_2, \ldots, \lambda_n)$ is the diagonal matrix consisting of the eigenvalues of $V$, and $P^*$ is the conjugate transpose of $P$.*

*Hence, we can rewrite*

$$\frac{ds_{t+k}}{dh_t} = \sum_{s=0}^{k} V^s \cdot \chi_{0:k}^{(t)}(s) = P \cdot \left( \sum_{s=0}^{k} \Lambda^s \cdot \chi_{0:k}^{(t)}(s) \right) \cdot P^* \tag{282}$$

*We can therefore see that the asymptotic behaviour of $\frac{ds_{t+k}}{dh_t}$ depends largely on the asymptotic behaviour of the modulus of the complex-valued polynomial*

$$p_{0:k}(\lambda) \overset{\text{def}}{=} \sum_{s=0}^{k} \lambda^s \cdot \chi_{0:k}^{(t)}(s) \tag{283}$$

*and thus*

$$\|\frac{ds_{t+k}}{dh_t}\| = \sqrt{\sum_{i=1}^{n} |p_{0:k}(\lambda_i)|^2} \tag{284}$$

*where $\|.\|$ is the Frobenius norm. Hence in order to prove that*

$$\|\frac{ds_{t+k}}{dh_t}\| = \Omega(1/\kappa^d) \tag{285}$$

*for all large enough $k$ (note that $k$ and $\kappa$ here are two different symbols), it would suffice to show that there exists $\lambda \in \{\lambda_1, \ldots, \lambda_n\}$ such that, for all large enough $k$, we have*

$$|p_{0:k}(\lambda)| = \Omega(1/\kappa^d) \tag{286}$$

*For simplicity we are going to assume that for all $t$, we have $\kappa_t = \kappa$.*

*Let us further define for all $s \geq 1$,*

$$f_{0:k}^{(s)}(i_1, \ldots, i_s) \overset{\text{def}}{=} \left( \frac{1_{t+i_s+1 \in A_{t+k}}}{\kappa_{t+k}} + 1_{k-i_s=1} \right) \cdot \left( \frac{1_{t+i_{s-1}+1 \in A_{t+i_s}}}{\kappa_{t+i_s}} + 1_{i_s-i_{s-1}=1} \right) \cdot \ldots \tag{287}$$

$$\ldots \cdot \left( \frac{1_{t+i_1+1 \in A_{t+i_2}}}{\kappa_{t+i_2}} + 1_{i_2-i_1=1} \right) \cdot \left( \frac{1_{t \in A_{t+i_1}}}{\kappa_{t+i_1}} + 1_{i_1=0} \right) \tag{288}$$

*whenever $(i_1, \ldots, i_n)$ satisfies $0 \leq i_1 < i_2 < \ldots < i_s < k$, and*

$$f_{0:k}^{(s)}(i_1, \ldots, i_s) \overset{\text{def}}{=} 0 \tag{289}$$

*otherwise.*

---

**Theorem 2.** *Given the $\kappa$-sparsity assumption and the dependency depth $d$, we have that if $V$ is normal and has one positive real eigenvalue, then*

$$\|\frac{ds_{t+k}}{dh_t}\| = \Omega(1/\kappa^d) \tag{290}$$

*for all large enough $k$.*

*Proof.* By the hypothesis on the dependency depth $d$, we know that for each $k$, there exists $s' \leq d$ and $(i_1, i_2, \ldots, i_{s'})$ such that

$$f_{0:k}^{(s')}(i_1, \ldots, i_{s'}) \geq \left(\frac{1}{\kappa}\right)^{s'+1} \geq \left(\frac{1}{\kappa}\right)^{d+1} \tag{291}$$

Hence if $\lambda$ is real and positive, then for all large enough $k$, we have

$$|p_{0:k}(\lambda)| = \Omega(1/\kappa^d) \tag{292}$$

Let us recall that, since $V$ is normal we can write

$$\|\frac{ds_{t+k}}{dh_t}\| = \sqrt{\sum_{i=1}^{n} |p_{0:k}(\lambda_i)|^2} \tag{293}$$

where $\lambda_1, \ldots, \lambda_n$ are the eigenvalues of $V$.

Hence, if $V$ has at least one positive real eigenvalue then

$$\|\frac{ds_{t+k}}{dh_t}\| = \Omega(1/\kappa^d) \tag{294}$$

for all large enough $k$. $\qquad\square$

---

**Remark 21.** *As already mentioned, since $\kappa$ and $d$ are assumed to be constant, the theorem states that*

$$\|\frac{ds_{t+k}}{dh_t}\| = \Omega(1) \tag{295}$$

*The dependence on $\kappa$ and $d$ was simply given in order to get an intuition on how $\kappa$ and $d$ are influencing the lower bound, and that $d$ has more leverage on the lower bound than $\kappa$.*

*Regarding the normality of $V$, the same remark can be made as in remark 17.*

*Then note that if $V$ is a (real) $n \times n$ matrix, with $n$ odd, then we have at least one real eigenvalue. Thus the restriction of having at least one positive real eigenvalue is not that severe.*

*Further, one can show that the theorem holds in a slightly more general setting where one might not have at least one positive real eigenvalue.*

*Let us consider the case where $\kappa = 1$, $|\lambda| < 1$ such that we could consider $\lambda^c \approx 0$ for some large enough positive integer $c$, and that all states between $T$ and $T - c$ have dependency depth of exactly $d$ (where $T = t + k$), then*

$$p_{0:k}(\lambda) = \frac{\lambda^d}{\kappa^d} \cdot (1 + \lambda + \ldots + \lambda^{c-d}) = \frac{\lambda^d}{\kappa^d} \cdot \left(\frac{1 - \lambda^{c-d+1}}{1 - \lambda}\right) \tag{296}$$

*Hence we can see that if we can show that $\left|\frac{1-\lambda^{c-d+1}}{1-\lambda}\right|$ is lower bounded asymptotically by a constant, independent of $d$ and $\kappa$, (which it is in this case), then we have*

$$|p_{0:k}(\lambda)| = \Omega(1/\kappa^d) \tag{297}$$

*We also see that we would like $\lambda$ to be sufficiently bounded away from a small set of critical values such as the $(c - d)$-th roots of unity.*

*In a more general setting, we can rewrite*

$$p_{0:k}(\lambda) = \frac{\lambda^d}{\kappa^d} \cdot q_{0:k}(\lambda) \tag{298}$$

*for some polynomial $q_{0:k}$ with positive real coefficients, and we would like $\lambda$ to be such that $|q_{0:k}(\lambda)| = \Omega(1)$ for all sufficiently large $k$.*

*Our hypothesis is that the theorem holds as long as $\lambda$ is sufficiently bounded away from a small set of critical values in $\mathbb{C} \setminus \mathbb{R}^+$, or in other words, we would need only at least one eigenvalue to satisfy this condition. This set of critical values is a dependent on $\kappa$, $d$ and the overall configuration of the attention weights.*

# B  Effects of memory sparsity on basic reinforcement learning tasks

We consider a few tasks from MiniGrid [8] in the OpenAI gym [6] in which an agent must get to certain goal states. We use a partially observed formulation of the task, where the agent only sees a small number of squares ahead of it. Our goal is to compare generalization of the solutions learned by full and sparse memory-augmented models, by training on smaller version of an environment and testing it on a larger version. To do so, we compare the use of MemLSTM (full attention) and RelLSTM (sparse attention). We note that some purely recurrent models can perform well on these tasks where sequence lengths are rather short, but the scope of this experiment is to explicitly compare the effect of different memory densities.

Table 4: Average Train and Test Rewards for MiniGrid Reinforcement Learning task. The models were trained on the smaller version of the environment and tested on the larger version to test to generalization of the solution learned.

| Environment | MemLSTM | RelLSTM |
|---|---|---|
| **Train** | | |
| RedBlueDoors-6x6 | **0.97** | **0.97** |
| GoToObject-6x6 | **0.85** | 0.84 |
| MemoryS7 | 0.4 | **0.94** |
| GoToDoor-5x5 | 0.17 | **0.25** |
| Fetch-5x5 | 0.42 | **0.5** |
| DoorKey-5x5 | **0.94** | 0.93 |
| **Test** | | |
| RedBlueDoors-8x8 | **0.95** | **0.95** |
| GoToObject-8x8 | 0.66 | **0.74** |
| MemoryS13 | 0.24 | **0.30** |
| GoToDoor-8x8 | 0.11 | **0.15** |
| Fetch-8x8 | 0.44 | **0.45** |
| DoorKey-16x16 | 0.31 | **0.44** |

These tasks are difficult to solve with standard RL algorithms, due to (1) the partial observability of the environment and (2) the sparsity of the reward, given that the agent receives a reward only after reaching the goal. We use Proximal Policy Optimization (PPO, [36]) along with MemLSTM, and RelLSTM as the recurrent modules. All models were each trained for 5000000 steps on each environment. The hyperparameters used for RelLSTM are $\nu = 5$ and $\rho = 5$. On the *MiniGrid-DoorKey-5x5-v0* environment the average reward for MemLSTM is $0.94$ and RelLSTM is $0.93$. On transferring the learned solution to the *16x16* version of that environment the average reward for MemLSTM is $0.31$ and RelLSTM is **0.44**. As illustrated in 4, we find that transfer scores for RelLSTM are much higher than for MemLSTM across several environments.

# C  Tradeoff analysis between sparsity and gradient propagation

As already discussed in Section 4, the sparsity coefficient $\kappa$ verifies $\kappa = \nu + \rho \geq |S_t| + |R_t|$ for all time step $t$, where we denote $\nu$ for the size of the short-term buffer, and $\rho$ for the maximal size of the relevant sets $R_t$. In this section we would like to see how gradient propagation varies when changing sparsity. As already discussed at the end of Section 3 as well as at the end of Section 4, decreasing $\kappa$, would increasingly force gradients to backpropagate through the recurrent connections, thus degrading gradient stability. Meanwhile, increasing $\kappa$ would increase the size of the computational graph. Thus we would like to find the optimal trade-off between sparsity and gradient propagation. This trade-off is clearly task-specific and needs to be determined experimentally. The only way to do so is by either changing $\nu$ or changing $\rho$ (or both). Hence we are going to analyze the effects on gradient propagation by separately changing $\nu$ and $\rho$.

Figure 3: Both sides show gradient norm plots of $\|\nabla_{h_t} L\|$ in log scale after training for Denoise Task with $t$ ranging from 0 (latest time step) to 1000 (furthest time step). **(Left)** We took four MemLSTM models for $\rho = 3, 8, 18, 25$ while keeping $\nu = 15$ fixed. **(Right)** We took four MemLSTM models for $\nu = 3, 8, 18, 25$ while keeping $\rho = 15$ fixed. (Note that the $y$-axis of the two plots have different scales, as indicated in the plots.)

For Figure 3 (left), we can see that when choosing $\rho$ too small (here for instance $\rho = 3$), gradient propagation becomes unstable, while larger values for $\rho$ all show stable gradient propagation. This confirms our initial intuition that we can decrease $\rho$ until a task-specific treshold and maintain stable gradient propagation, while decreasing $\rho$ beyond this treshold would cause gradient propagation to become unstable.

For Figure 3 (right), we can see that changing $\nu$ has much less leverage on gradient propagation than changing $\rho$. Gradient propagation stays relatively stable regardless of the values for $\nu$. The only difference is that for the extreme value of $\nu = 3$, we can see that gradient propagation became slightly less stable, because with smaller $\nu$ predictions for future relevancy might become less accurate.

# D Additional Results

Figure 4: Cross-entropy vs training updates for Copy (top) and Denoise (bottom) tasks for $T = \{100, 200, 300, 500, 1000, 2000\}$. 1 unit of the x-axis is equal to 100 iterations of training with the exception of expRNN where 1 unit on the x-axis is 10 iterations of training.

Table 5: Results for Copy Task

| $T$ | LSTM | orth-RNN | expRNN | MemRNN | SAB | RelRNN | RelLSTM |
|------|------|----------|--------|--------|------|--------|---------|
| 100  | 100% | 100%     | 100%   | 100%   | 100% | 100%   | 100%    |
| 200  | 100% | 100%     | 100%   | 100%   | 100% | 100%   | 100%    |
| 300  | 100% | 100%     | 100%   | 100%   | 100% | 100%   | 100%    |
| 500  | 12%  | 100%     | 100%   | 100%   | 100% | 100%   | 100%    |
| 1000 | 12%  | 80%      | 100%   | 100%   | 100% | 100%   | 100%    |
| 2000 | 12%  | 11%      | 100%   | OOM    | 100% | 100%   | 100%    |

Table 6: Hyperparameters used for Copy task

| Model   | lr     | optimizer | non-linearity | $\nu$ | $\rho$ |
|---------|--------|-----------|---------------|-------|--------|
| orthRNN | 0.0002 | RMSprop   | modrelu       | -     | -      |
| expRNN  | 0.0002 | RMSprop   | modrelu       | -     | -      |
| LSTM    | 0.0002 | Adam      | -             | -     | -      |
| RelRNN  | 0.0002 | Adam      | tanh          | 10    | 10     |

Figure 5: Training curves for LSTM on Denoise task

Figure 6: Training curves for GORU on Denoise task

Figure 7: Heatmap of attention scores on PTB task training with full attention and BPTT of 125

Figure 8: Validation accuracy curves for pMNIST

Figure 9: Heatmap of attention scores on MNIST digit classification. 7 pixels were grouped at each time step to make visualization of heatmap easier.

Figure 10: Heatmap of attention scores on Copy task when only doing attention over the Short Term Buffer.

Table 7: Hyperparameters used for Denoise task

| Model | lr | optimizer | non-linearity | $\nu$ | $\rho$ |
|---|---|---|---|---|---|
| orthRNN | 0.0002 | RMSprop | modrelu | - | - |
| expRNN | 0.0002 | RMSprop | modrelu | - | - |
| LSTM | 0.0002 | Adam | - | - | - |
| GORU | 0.001 | RMSprop | - | - | - |
| RelRNN | 0.0002 | RMSprop | modrelu | 10 | 10 |

Table 8: Hyperparameters used for sequential MNIST

| Model | lr (lr orth) | optimizer | non-linearity | $\nu$ | $\rho$ |
|---|---|---|---|---|---|
| orthRNN | 0.0001 | Adam | modrelu | - | - |
| expRNN | 0.0001(0.00001) | Adam | modrelu | - | - |
| LSTM | 0.0002 | | - | - | - |
| GORU | | | - | - | |
| RelRNN | 0.0003 | Adam | modrelu | 10 | 10 |

Figure 11: Heatmap of attention scores on Denoise task when only doing attention over the Short Term Buffer.

Table 9: Hyperparameters used for PTB

| Model | lr (lr orth) | optimizer | non-linearity | $\nu$ | $\rho$ |
|---|---|---|---|---|---|
| orthRNN | 0.001 | Adam | tanh | - | - |
| expRNN | 0.003(0.0003) | Adam | tanh | - | - |
| LSTM | 0.0002 | | - | - | - |
| GORU | | | - | - | |
| RelRNN | 0.0003 | Adam | tanh | 10 | 5 |