[Reviews · NeurIPS 2020]

Review 1

Summary and Contributions: Thanks to the authors for an informative rebuttal. The rebuttal addressed my concerns to a satisfactory level: so I'd like to increase my evaluation by +1 and think this paper should be accepted. For the ``faster'' comment, I strongly encourage the authors to discuss not only the # of updates but also the actual wall-clock time (which I believe should be generally proportional to the # of updates?). I am sorry that I didn't fully appreciate the carefully documented appendices in my initial review. For the final version, I strongly recommend that the authors refer to the appendices for important remarks to make in the main paper. ++++++ The paper theoretically analyzes how self-attention helps mitigate the problem of gradient vanishing in training an RNN and proposes a screening mechanism that enables sparse self-attention that doesn't suffer quadratically increasing memory.

Strengths: The work provides sound theoretical analysis on how self-attention can help resolve the gradient vanishing problem and how sparse self-attention can prevent gradient vanishing while still maintaining constant memory usage. Full proofs with extensive details are provided in the supplementary material. The proposed screening mechanism turns out to be effective on several tasks, achieving compelling results while still being fast and memory-efficient. Empirical analysis also demonstrates that the gradient, using this mechanism, doesn't explode or vanish.

Weaknesses: The authors didn't spell out the relation between \kappa and d: higher \kappa tends to have smaller d. This relation is discussed in the paragraph of line-173 but not reflected in the formula of theorem-2. In experiments, the authors mentioned the proposed model is ``faster'' to train but didn't give any quantitative results. It seems useful to also show such quantitative results just as they do for the memory usage analysis.

Correctness: The theorems and proofs look correct to me. The method looks reasonable.

Clarity: The paper is overall well written, except that the mathematical notations seem a bit messy. E.g., the letter s is overloaded many times which makes it hard to follow---s can be neural state (eqn-2), index of i (proposition-1), depth s (definitions), etc. E.g., BPC in table-2 is undefined.

Relation to Prior Work: The paper clearly discusses how it is related to previous contributions.

Reproducibility: Yes

Additional Feedback:


Review 2

Summary and Contributions: This paper presents a formal treatment of gradient propagation in attentive recurrent neural networks, in which the gradient norm is asymptotically quantified based on attention sparsity (k) and maximal dependency length (d). To balance between computational complexity (small k) and good gradient propagation (small d), a relevancy screening mechanism is introduced, retaining only top relevant timesteps for attention. The method is tested on synthetic memorization tasks, Penn Tree Bank, and MiniGrid reinforcement learning environment, demonstrating promising results, especially on generalization capability.

Strengths: - The theoretical analysis is interesting, providing a solid foundation for future theoretical works in attention mechanism - The method is simple, yet effective for the considering tasks

Weaknesses: - The literature review is incomplete, lacking detailed comparisons to relevant prior works - The empirical results are not strong - The method is only verified on simple/synthetic tasks

Correctness: Seems correct. I have not checked the theory clearly

Clarity: Overall yes

Relation to Prior Work: Not really

Reproducibility: Yes

Additional Feedback: - Line 145, how can Theorem 1 be related to the early attention mechanism [1]? As the attention weights are computed adaptively, it is unlikely that they are uniform. - Memory-augmented neural networks (MANN [2,3]) are naturally sparse self-attentive RNNs (k is the number of memory slots). MANNs learn to store relevant hidden states to a fixed-size memory, which seems to have the same purpose as relevancy screening mechanism. What is the advantage of the proposed method over MANNs? How are MANNs related to the Theorem 2? - The paper neglects prior works that also aim to quantify gradient propagation in RNNs and attentive models [4,5]. How are the paper’s theoretical findings different from [5], wherein gradient norm of self-attentive RNN is also quantified? A clear comparison is necessary to highlight the theoretical contribution of the paper. - Using attention weights as a relevant indicator is explored before under reinforcement learning context [6]. A clear comparison is necessary to highlight the novelty of the proposed relevancy screening mechanism. - Fig.1, values on the axis are too small - Line 260, Table 4 is not about Copy task - Table 3 is confusing. For example, in pMNIST, relLSTM underperforms expRNN yet both are bold. - In experiments, the authors should include stronger baselines to prove the benefit of the approach. For example, MANNs in [5] performs well on pMNIST tasks. In RL task, it is better to have a strong baseline such as [6]. [1] Dzmitry Bahdanau, Kyunghyun Cho, and Yoshua Bengio. Neural Machine Translation by Jointly Learning to Align and Translate. arXiv e-prints, art. arXiv:1409.0473, Sep 2014. [2] Alex Graves, Greg Wayne, and Ivo Danihelka. Neural turing machines. arXiv preprint arXiv:1410.5401, 2014. [3] Alex Graves, Greg Wayne, Malcolm Reynolds, Tim Harley, Ivo Danihelka, Agnieszka Grabska- Barwin ́ska, Sergio Gómez Colmenarejo, Edward Grefenstette, Tiago Ramalho, John Agapiou, et al. Hybrid computing using a neural network with dynamic external memory. Nature, 538 (7626):471, 2016. [4] Shiyu Chang, Yang Zhang, Wei Han, Mo Yu, Xiaoxiao Guo, Wei Tan, Xiaodong Cui, Michael Witbrock, Mark A. Hasegawa-Johnson, and Thomas S. Huang. "Dilated recurrent neural networks." In Advances in Neural Information Processing Systems, pp. 77-87. 2017. [5] Hung Le, Truyen Tran, and Svetha Venkatesh. "Learning to remember more with less memorization." arXiv preprint arXiv:1901.01347 (2019). [6] Chia-Chun Hung, Timothy Lillicrap, Josh Abramson, Yan Wu, Mehdi Mirza, Federico Carnevale, Arun Ahuja, and Greg Wayne. "Optimizing agent behavior over long time scales by transporting value." Nature communications 10, no. 1 (2019): 1-12. ========= Update after rebuttal: While I feel satisfied with the authors' response regarding the theoretical contribution of the paper, I find the algorithm and experiments less convincing. The authors haven't answered my question on the novelty of the proposed relevancy screening mechanism. Hence, I would like to keep my score as is.


Review 3

Summary and Contributions: The paper tackle the gradient vanishing problem of RNNs by augmenting with a sparse attention mechanism that supported by a theoretical analysis. The sparsity is achieved by prioritizing memories that attended more often during previous time steps. The model is compared to other RNN based baselines on toy tasks, language modeling, and pMNIST.

Strengths: The paper is well motivated and that motivation is backed by a detailed theoretical analysis. The theoretical part clearly shows the benefit of attention in RNNs, but also go further by showing that sparse attention work better than dense one. In addition, the paper proposed a novel sparse attention mechanism inspired by brain that is simple to implement. More importantly, the method reduces both computational time and memory usage as demonstrated by the experiments.

Weaknesses: I have few minor comments about the novelty and the theory in the paper, but my main concern is with the experiments. - Novelty: augmenting RNNs with sparse attention to prevent gradient vanishing is not novel in itself [19]. The only novel part of the model is relevancy screening, but why that approach is better than other sparsity methods has no grounding in the theoretical analysis in the paper. Actually, the theoretical analysis itself is based on the top-k approach from [19] instead of relevancy screening. - Theoretical part: it is well known that an attention mechanism would reduce gradient vanishing. It feels trivial to me as there is a direct connection for gradients to pass. As that connection is weighted by softmax attention weights that sum to one, it's not hard to see that having fewer things in the softmax (i.e. sparse attention) would increase the attention weights, thus improved gradient flow. - Experiments: I think the experiments in the paper are quite weak. The only tasks that are not toy are char-PTB and pMNIST. But even char-PTB is small when compared to commonly used LM benchmarks like text8, enwik8, wikitext103. In addition, there was no improvement over a vanilla LSTM on char-PTB. The improvement in pMNIST is also marginal. And this is without including Transformer based methods, which work much better than RNNs on such tasks.

Correctness: The theoretical claims make sense and seem correct to me, but I didn't check the proofs in the supplementary material. The experimental setups also look good to me.

Clarity: In general, the paper is well written and easy to understand. Minor comments: - There was not much about the proposed method in the introduction. Explaining "relevancy screening" a bit more would make it easier to understand in a limited time. - I think the brain related claims like "just like NN in the brain" , "our brain do ... attempts at mimicking this ..." can be bit toned-down as they're only inspirations. - Typo: Fig 2 is referred as Fig 6 in lines 319-324.

Relation to Prior Work: Yes. I just have few minor comments: - The method has a lot of similarity with [19], but when discussing [19] in the background section, the authors only said "naively sub-sample input streams for memory usage", which is too vague and unclear to me. The top-k sparsity from [19] is simple, but works well and used in many subsequent works. Instead, I think the authors can mention the computational complexity of [19] as discussed in Sec. 4 and 6. - It is true that self-attention has a quadratic complexity, but there is a simple way to make it linear by limiting the attention span to recent L tokens as done in Transformer-XL, which is worth mentioning in the background section.

Reproducibility: Yes

Additional Feedback: === Post-rebuttal comment: After reading the rebuttal, I'm upgrading my score to 7.

[Author Response · NeurIPS 2020]

We thank the reviewers for insightful remarks and comments that help to considerably improve our manuscript. We address the most important ones in detail below. Before doing so, we highlight a comment from R3 in order to make an important clarification about the scope of our contribution. "*It is well known that an attention mechanism would reduce gradient vanishing. It feels trivial to me as there is a direct connection for gradients to pass. [...] it's not hard to see that having fewer things in the softmax (i.e. sparse attention) would increase the attention weights, thus improved gradient flow.*" We are in complete agreement and recognize that the very mechanism of (self-)attention was designed to improve gradient propagation over long sequences, and that sparsity is a good way to keep complexity costs low. However, to the best of our knowledge, there is currently limited understanding of gradient scaling properties in the presence of attention. Much like work from the '90s established formal results for gradient exploding/vanishing in deep/recurrent networks, we believe it is crucial to establish similar theoretical tools for attention mechanisms, as these methods are under intense development where scalability and complexity are important issues. Our main aim is to contribute to this direction with a thorough analysis of gradient propagation in self-attentive systems, outlined in clear theorem statements which precisely quantify the intuition expressed by R3. These results are not trivial (see proofs in appendices), and offer valuable guarantees for attention mechanism development. The proposed relevancy mechanism and accompanying experiments, building on established work, are meant to illustrate how our theorems can be concretely exploited. We chose simple tasks for their ease of interpretation, and their variety of computational demands (memorization, prediction, RL, etc.). As is clearly indicated in the text, it is not our goal to propose this method "as is" in a race for state-of-the-art. Rather, we think it achieves its goal of providing a firm theoretical footing for a wide class of self-attention methods as well as suggesting future direction of methods based on concrete scaling guarantees and a balance between sparsity and attention complexity. We recognize that reviewers may have based their evaluation as they would have in a method paper, and we kindly invite them to reconsider the value of our experiments in the broader context of our theoretical contributions. We also thank reviewers for their additional minor comments not explicitly addressed here and agree to implement them.

**R1: Q** "*The authors didn't spell out the relation between $\kappa$ and $d$: higher $\kappa$ tends to have smaller $d$. This relation is discussed in the paragraph of line-173 but not reflected in the formula of theorem-2.*" **A:** We thank R1 for this important remark. As it stands, Theorem 2 offers scaling relationship for any $\kappa$ and $d$. However in practice, $\kappa$ and $d$ co-vary in ways that depend on the task's underlying relevancy structure, a point that is explained in detail in Appendix C (see Fig 3) which explores trade-offs between $\rho$ $(= \kappa - \nu)$ and gradient propagation (implicitly depending on $d$). We will sharpen this discussion in the main text. **Q** "*In experiments, the authors mentioned the proposed model is "faster" to train but didn't give any quantitative results..*" **A:** We will clarify this in the text and specifically point to Fig 4 in the Appendix D (or move it to the main text space permitting) which shows RelLSTM/RelRNN learns the copy and denoise tasks with significantly fewer number of updates as compared to other baselines.

**R2: Q** "*Line 145, how can Theorem 1 be related to the early attention mechanism [1]? As the attention weights are computed adaptively, it is unlikely that they are uniform.*" **A:** We recognize that uniform attention weights are unlikely in practice. This theorem offers a form of "worst case" guarantees which is applicable in practice for two reasons. (1) Typically, attention weights are initialized uniformally. (2) We experimentally verified that gradient propagation remains stable throughout training for a fully self-attentive RNN, see Fig 2 (Section 6). We will further clarify this in the text. **Q** "*What is the advantage of the proposed method over MANNs? / how are MANNs related to the Theorem 2? How are the paper's theoretical findings different from [5], wherein gradient norm of self-attentive RNN is also quantified?*" **A:** We thank R2 for this insightful remark and acknowledge the lack of discussion about the MANN model class. Most instances operate on involved memory addressing/retrieval mechanisms that keep complexity costs low but, to our knowledge, do not offer gradient propagation guarantees like Theorem 2. Some instances, such as that proposed in [5], offer more similarities to our approach, and future adaptations of our Theorem 2 is an exciting future direction. Our contribution complements that of [5] in two subtle but important ways. (1) The proof of Theorem 2 in [5] describes an approximation of information propagation via recurrence, with "skip connections" contributions accounted only once in isolation. In self-attention, gradient propagates via a mix of skip and recurrent connections, leading to multiple gradient paths. Enumerating these paths and incorporating their contributions to gradient norms is the crux of our results, yielding complete and precise expansions of gradient terms applicable to a wide range of models, including Transformers. (2) The proposed method in [5] is an optimal memory writing schedule. In contrast, our method relies on relevancy for memory writing, allowing more flexibility in complexity scaling. We will incorporate this discussion in the revised text.

**R3: Q** "*augmenting RNNs with sparse attention to prevent gradient vanishing is not novel in itself [19]. The only novel part of the model is relevancy screening, but why that approach is better than other sparsity methods has no grounding in the theoretical analysis in the paper [...].*" **A:** We agree with R3 that sparse attention is not novel. However, quantifying how much sparsity contributes to gradient norm is new. The theorems we provide borrow notation from [19], but apply to a large class of self-attentive mechanisms as well. To see why theorem 2 directly applies to the relevancy screening mechanism, it suffices to take $\kappa = \rho + \nu$, as at each time step we are attending to the $\rho$ states from the relevant set and the $\nu$ states from the short term buffer. To highlight how our relevancy screening is grounded in the theory, see Fig 3 in Appendix C, where we perform an experimental trade-off analysis between $\kappa$ and $d$ by tweaking $\rho$ and $\nu$ for our relevancy screening mechanism. This will be clarified in the text. We would like to emphasize that theorem 2 applies to any $\kappa$-sparse self-attention recurrent model, but we harness the structural understanding derived from the theoretical framework to propose a more scalable form of sparse self-attentive RNN.

**Q** "*Experiments: I think the experiments in the paper are quite weak [...].* **A:** We appreciate this remark and refer to this rebuttal's opening statement about scope. We also point out that our methods appreciably improves generalization to out-of-distribution samples over baseline. This is a promising avenue and will be further discussed in the text. We also want to thank R3 for pointing out the Transformer-XL reference, which we will make sure to include in the main text.



[Meta-Review · NeurIPS 2020]

The paper provides theoretical analysis of self-attention and vanishing gradients. Experiments are of toy problems with non-SOTA results but validate the main theoretical contributions of the paper.